

# pH-Dependent production of molecular chlorine, bromine, and
# iodine from frozen saline surfaces
John W. Halfacre[1], Paul B. Shepson[2,3], Kerri A. Pratt[4]
[1]Department of Chemistry, Indiana University Southeast, New Albany, IN USA
[2]Department of Chemistry, Purdue University, West Lafayette, IN USA
[3]Department of Earth, Atmospheric, and Planetary Sciences, Purdue University, West Lafayette, IN USA
[4]Department of Chemistry and Earth & Environmental Sciences, University of Michigan, Ann Arbor, MI USA
*Correspondence to:* J. W. Halfacre (halfacre@ius.edu)



**Abstract**

The mechanisms of molecular halogen productions from frozen saline surfaces remain incompletely

understood, limiting our ability to predict atmospheric oxidation and composition in the polar regions.  In this
laboratory study, condensed-phase hydroxyl radicals (OH) were photochemically generated in frozen saltwater
solutions that mimicked the ionic composition of ocean water. These hydroxyl radicals were found to oxidize $Cl^-$,
$Br^-$, and $I^-$, leading to the release of $Cl_2$, $Br_2$, and $I_2$.  This finding is consistent with mechanisms proposed from recent
Arctic field studies for observed snowpack molecular halogen production.  At moderately acidic pre-freezing pH
(buffered between 4.5-4.8), irradiation of ice surfaces containing OH-precursors produced elevated amounts of $I_2$.
Subsequent addition of $O_3$ produced additional $I_2$, as well as low amounts of $Br_2$.  At lower pH (1.7-2.2), substantial
photochemical production of $Br_2$ was observed, following rapid dark conversion of $I^-$ to $I_2$ via reactions with hydrogen
peroxide or nitrite.  Exposure to $O_3$ under these low pH conditions also increased production of $Br_2$ and $I_2$, possibly
through surfaced-based reactions with $O_3$, or the production and heterogeneous recycling of gas-phase HOBr and HOI.
Our results suggest the observed products are dependent on the relative concentrations of halides at the ice surface.
Finally, photochemical production of $Cl_2$ was only observed when the irradiated sample was composed of high-purity
NaCl and hydrogen peroxide (acting as the OH precursor) at low pH (~1.8).  While OH was shown to produce $Cl_2$ in
this study, kinetics calculations suggest that heterogeneous recycling chemistry may be equally or more important for
$Cl_2$ production in the Arctic atmosphere.





## 1 Introduction

It is now well-established that gas-phase halogen species influence atmospheric composition through reactions with ozone ($O_3$), volatile organic compounds (VOCs), and gaseous elemental mercury ($Hg^0$) (Barrie and Platt, 1997; Carpenter et al., 2013; Platt and Hönninger, 2003; Saiz-Lopez and von Glasow, 2012; Simpson et al., 2007, 2015; Steffen et al., 2008, 2014, and references therein). In polar regions, it is believed that halogens build up to effective concentrations through a heterogeneous reaction sequence known as the "halogen explosion" (Reactions R1-4, where X represents Cl, Br, or I) (Garland and Curtis, 1981; Tang and McConnell, 1996; Vogt et al., 1996; Wennberg, 1999).

$$X_2 + h\nu \rightarrow 2X \tag{R1}$$

$$X + O_3 \rightarrow XO + O_2 \tag{R2}$$

$$XO + HO_2 \rightarrow HOX + O_2 \tag{R3}$$

$$HOX + X^- + H^+ \rightarrow X_2 + H_2O \tag{R4}$$

In this sequence, a molecular halogen ($X_2$) is photolyzed to produce two reactive halogen radicals. These radicals can react with $O_3$ to produce halogen oxides. XO produced in Reaction R2 rapidly photolyzes (or reacts with NO) to regenerate $O_3$ and $X_2$ in a null cycle. To irreversibly remove ambient $O_3$, XO must react with another halogen oxide or $Hg^o$. Alternatively, XO can react with $HO_2$ to form HOX (Reaction R3) or $NO_2$ to form $XONO_2$. Gas-phase HOX can heterogeneously react with salt-laden surfaces, including sea-salt aerosol particles (McConnell et al., 1992) and the "disordered interface" (often referred to as a quasi-liquid or quasi-brine layer) that exists on frozen saline surfaces (Bartels-Rausch et al., 2014; Cho et al., 2002) to produce $X_2$, effectively returning two halogen radicals to the gas phase. Additionally, this mechanism is enhanced under acidic conditions, confirmed by laboratory studies of aqueous (Fickert et al., 1999) and frozen solutions (e.g., Abbatt et al., 2010; Sjostedt and Abbatt, 2008; Wren et al., 2013), and from field observations (Pratt et al., 2013).

While much has been learned about the atmospheric chemistry of reactive halogen species in the Arctic, knowledge gaps remain in the chemical mechanisms by which molecular halogens are produced on frozen surfaces (Liao et al., 2014; Pratt et al., 2013). However, there have been recent reports of in situ, light-induced production of $Cl_2$ (Custard et al., 2016), $Br_2$ (Pratt et al., 2013; Raso et al., 2017), and $I_2$ (Raso et al., 2017) within snowpack interstitial air, and further suggest that this production is enhanced following the addition of $O_3$. The $Br_2$-producing snowpacks studied by Pratt et al. (2013) were characterized as having larger surface area, lower pH ($\leq$ 6.3), greater





[Br⁻]/[Cl⁻] molar ratios (≥ 1/148), and lower salinity relative to other frozen samples collected near Utqiaġvik, Alaska.
The proposed mechanism for this chemistry is based on laboratory studies of condensed-phase, hydroxyl radical (OH)-
mediated halogen oxidation (Reactions R5-R12), followed by partitioning of the molecular halogen to the gas phase
(Abbatt et al., 2010; Knipping et al., 2000; Oum et al., 1998b).
$H_2O_2 + h\nu \rightarrow 2\ OH$                                                                                                     (R5)
$NO_2^- + h\nu \rightarrow NO + O^-$                                                                                              (R6)
$O^- + H^+ \rightarrow OH$                                                                                                              (R7)
$OH + X^- \leftrightarrow HOX^-$                                                                                                      (R8)
$HOX^- + H^+ \rightarrow X + H_2O$                                                                                              (R9)
$X + X^- \leftrightarrow X_2^-$                                                                                                          (R10)
$2X_2^- \rightarrow X_3^- + X^-$                                                                                                      (R11)
$X_3^- \leftrightarrow X^- + X_2$                                                                                                      (R12)
Direct, light-induced halogen production from frozen surfaces in the presence of OH has been previously demonstrated
in the laboratory for $Br_2$ and possibly for $I_2$ (Abbatt et al., 2010), but analogous chemistry for $Cl_2$ has yet to be observed.
Additionally, photochemical production of $I_2$ has been directly observed in the absence of OH (Kim et al., 2016).
Employing cavity ring-down spectroscopy, Kim et al. (2016) reported photochemical production of $I_2$ from a frozen
solution by known aqueous-phase chemistry (R13-17).  This proposed photochemical mechanism involves an ($I^-$-$O_2$)
charge-transfer complex (Levanon and Navon, 1969).
$O_2(aq) + 4H^+ + 6I^- \rightarrow 2I_3^- + 2H_2O$                                                                  (R13)
$I^- + O_2 \rightarrow (I^-O_2) \xrightarrow{h\nu} I + O_2^-$                                                          (R14)
$I + I^- \leftrightarrow I_2^-$                                                                                                              (R15)
$2I_2^- \rightarrow I_3^- + I^-$                                                                                                          (R16)
$I_3^- \leftrightarrow I^- + I_2$                                                                                                          (R17)
Kim et al. (2016) also report enhanced photochemical $I_3^-$ production (determined spectrophotometrically) from sunlit
frozen iodide solutions placed on Antarctic snowpack, as well as from refrozen field snow and glacier samples doped



with iodide.  A question is thus raised regarding the necessity of OH for $I_2$ production under environmentally-relevant
conditions.

The role of $O_3$ in halogen production on frozen surfaces is also unclear.  Previous laboratory studies have

demonstrated that halide-doped frozen surfaces exposed to $O_3$ can lead to $Br_2$ production (independent of radiation,
R18-R19, and R4) (Oldridge and Abbatt, 2011; Oum et al., 1998a; Wren et al., 2013).
$O_3 + Br^- \leftrightarrow BrO^- + O_2$                                                              (R18)
$BrO^- + H^+ \leftrightarrow HOBr$                                                             (R19)
It has recently been shown that this process proceeds at the surface, through a water-stabilized ozonide, $Br\cdot OOO^-$, as
shown in reactions R20-R22.  Artiglia et al. (2017) observed this $Br\cdot OOO^-$ intermediate via liquid-injection X-ray
photoelectron spectroscopy.
$Br^- + O_3 \rightarrow Br\cdot OOO^-$                                                      (R20)
$Br\cdot OOO^- + H^+ \rightarrow HOBr + O_2$                                         (R21)
$Br\cdot OOO^- + H_2O \rightarrow HOBr + O_2 + OH^-$                             (R22)
Wren et al. (2013) found that $Cl_2$ was produced primarily via heterogeneous recycling of HOCl, resulting from BrCl
photolysis, on halide-rich artificial snow.  However, the observation that $O_3$ induces halogen production from frozen
surfaces has yet to be confirmed by field observations of snowpack chemistry, in which exposure to only $O_3$ in the
absence of light has not been shown to produce molecular halogens.  Such in-situ experiments in the presence of $O_3$
were shown to produce enhanced molecular halogens only in the presence of light (Custard et al., 2017; Pratt et al.,
2013; Raso et al., 2017), raising a question of whether $O_3$ is more important for initial halogen release, or in a gas
phase propagation/recycling capacity (i.e., per the halogen explosion).

In this study, we utilized a custom ice-coated-wall flow reactor in tandem with chemical ionization mass

spectrometry (CIMS) to study $Br_2$, $Cl_2$, and $I_2$ production from frozen surfaces with compositions mimicking sea ice.
The effects of photochemically generated OH radicals, $O_3$ addition, and pH are tested as they relate to the production
of these halogens.  Surface pH was controlled through use of buffers.



## 2 Methods

### 2.1 Materials



Sample solutions were composed to mimic the halide composition of seawater. This was achieved using
either dissolved Instant Ocean (Spectrum Brands), or commercially available halide salts at a composition that mimics
Instant Ocean (for consistency) in solutions referred to hereafter as "saltwater." The halide concentrations in these
solutions were made to a final concentration of 0.56M $Cl^-$, 7.2 x $10^{-4}$ M $Br^-$, and 1.9 x $10^{-6}$ M $I^-$. Except for Instant
Ocean, all chemicals were purchased from Sigma Aldrich. Halide salts include solid NaCl (puriss. p.a. grade, ≥99.5%
purity), NaBr (puriss. grade, >99% purity), KI (puriss. p.a. grade, ≥99.5% purity). Sigma Aldrich lists an $I^-$ impurity
in the NaCl salt of ≤ 10 mg/kg, but the initial aqueous concentration of $I^-$ in this solution could not be confirmed by
ion chromatography. We note that these halide concentrations are comparable to those in actual seawater (Herring
and Liss, 1974; Luther et al., 1988; Tsunogai and Sase, 1969), which typically contains $Cl^-$, $Br^-$, and $I^-$ at ratios of
$1:^1/_{660}:^1/_{200,000}$. Solutes were dissolved in ultrapure water (Birck Nanotechnology Center). Dissolved organic carbon
for Instant Ocean and halide salt solutions were analyzed using a Shimadzu TOC-$V_{CSH}$ Total Organic Carbon
Analyzer, and determined at approximately 70 mg/L for Instant Ocean solutions, and less than 5 mg/L for saltwater
solutions. No further characterization of carbon-containing compounds was performed.
While previous investigators have adjusted the pre-freezing pH of their samples, it is very difficult to know
the pH in the disordered interface of frozen samples (Bartels-Rausch et al., 2014), though there is evidence from
laboratory studies suggesting that the pH of salt solutions remains largely unchanged after freezing (Wren and
Donaldson, 2012b). To obviate this problem, the aqueous solutions used in this study were buffered so that the same
pH will exist in the frozen disordered interface. All solutions were buffered by either 20 mM acetic acid (ACS reagent
grade, ≥99.7% purity)/acetate (puriss. p.a. grade) buffer (pH ≈ 4.5-4.7), or 20 mM bisulfate (ReagentPlus grade, 99%
purity)/sulfate (ReagentPlus grade, ≥ 99.0% purity) buffer (pH ≈1.7 – 2.0). pH values of sample solutions were
determined before and after experiments, with no significant changes observed. 100 µM of either hydrogen peroxide
(trace analysis grade, ≥ 30% purity) or sodium nitrite (ReagentPlus grade, ≥ 99.0% purity) were included as
photochemical hydroxyl radical precursors, via reactions R5, and R6-7.



## 2.2 Flow tube

Experiments were carried out in a custom-built 150 cm long, 2.5 cm ID frozen-walled Pyrex flow tube contained within a temperature-controlled cooling jacket. In each experiment, 80.0 mL of sample solution was poured into the tube, which was subsequently sealed with vinyl caps (McMaster-Carr). The flow tube was then rotated on motorized rollers within a 170 cm x 50 cm x 50 cm, insulated wooden cooling chamber. Crushed dry ice was placed along the bottom of the chamber, and fans were used to circulate the air throughout the chamber such that the flow tube was evenly cooled. After ~30 minutes, the sample was evenly frozen (ice thickness of 0.9 mm). The flow tube was subsequently transferred to an enclosed 156 cm x 50 cm x 50 cm wooden, Mylar-lined experiment chamber, and connected to a recycling chiller set to 258 K (i.e., above the NaCl·2H$_2$O eutectic point). The cooling liquid used for the chiller was a mixture of 60% ethylene glycol and 40% distilled water. Six UVA-340 solar simulator lamps (Q-Labs, 295 – 400 nm with maximum wattage at 340 nm) were installed in the experiment box (two on each side except bottom). Each side was lined with reflective Mylar sheets to evenly irradiate the flow tube when the lamps were powered.

A flow schematic representing typical experiments is shown in Fig. 1. The carrier gas (Air, Ultra Zero grade, Praxair) was scrubbed of volatile organic compounds using activated charcoal, and water by travelling through coiled stainless-steel tubing surrounded by crushed dry ice (replaced throughout the course of an experiment). This gas was measured to contain $\leq$ 300 – 400 pmol mol$^{-1}$ NO (experimentally determined limits of detection) using the Total REactive Nitrogen Instrument (TRENI) (Lockwood et al., 2010; Xiong et al., 2015). Though NO$_2$ was not measured, we believe it would have been removed by the charcoal trap. Before entering the experiment coated-wall flow tube, the carrier gas flowed through a commercial O$_3$ generator (2B Technologies model 306). Carrier gas air entered the tube near room temperature (20 ºC). At the start of experiments, the O$_3$ generator was set to 0 nmol mol$^{-1}$. Carrier gas then entered the flow tube in the dark experiment chamber. In most experiments, the carrier gas was regulated to a volumetric flow rate of 4.0 L min$^{-1}$, which yields a residence time in the flowtube of ~12 seconds. On exiting the flow tube, sample air was characterized using a Thermo Environmental 49i O$_3$ monitor (flow rate of ~1.5 L min$^{-1}$) and a chemical ionization mass spectrometer (CIMS, sampling flow rate of ~1.7 L min$^{-1}$, described below in Sect 2.3). Excess flow air was vented away. At set times in an experiment, the solar simulator bulbs were activated, and O$_3$ was added to the system by powering the O$_3$ generator.





### 2.3 CIMS


Halogen species were detected using a chemical ionization mass spectrometer (CIMS), described previously
by Liao et al. (2011) and Pratt et al. (2013). Chemical ionization is achieved by ion-molecule reactions that occur
between iodide-water reagent clusters, $I(H_2O)_n^-$ in $N_2$, and the gas-phase sample in zero air. The iodide-water clusters
are formed when gas-phase iodide ions, generated by flowing 5 ppm methyl iodide through a $^{210}Po$ ionizer (NRD)
combines with water in the humidified ion-molecule region of the CIMS. Ion were filtered using a quadrupole mass
filter. The ice-coated flowtube was connected to the CIMS via approximately 50 cm of i.d. 1/2″ PFA Teflon tubing.
A typical CIMS sampling cycle consisted of an 8.35s duty cycle. Dwell times for all monitored species were
250 ms with the exception of the reagent ion (detected as $m/z$ 147, $I(H_2^{18}O)^-$), which was set to a dwell time of 100
ms. The 18 ions analyzed in this study are listed in Table 1, but we focus herein on results concerning masses related
to $Br_2$ ($m/z$ 285 and 287: $I^{79}Br^{79}Br^-$ and $I^{81}Br^{79}Br^-$, respectively), $Cl_2$ ($m/z$ 197 and 199: $I^{35}Cl^{35}Cl^-$ and $I^{37}Cl^{35}Cl^-$), and
$I_2$ ($m/z$ 381: $I_3^-$). In addition, IBr ($m/z$ 333 and 335: $I^{79}IBr^-$, $I^{81}IBr^-$) was unambiguously detected in some experiments.
The presence of $Br_2$, $Cl_2$, and IBr was confirmed by measuring the ratios between the two isotope signals for each
mass, compared to the natural abundances (i.e., 1.95 for $m/z$ 287:285; 1.54 for $m/z$ 197:199; and 1.03 for $m/z$ 333:335,
respectively). Data outside ±25% the appropriate isotope ratio were excluded from analysis. The isotope ratios for
BrCl ($m/z$ 241 and 243: $I^{79}Br^{35}Cl^-$, $I^{81}Br^{35}Cl^-$, $I^{79}Br^{37}Cl^-$) masses were never observed at the correct values (1.3 for $m/z$
243:241), and so those data were not reported here. As the introduction of ~60 nmol mol$^{-1}$ $O_3$ to the experimental
system significantly increases the background signal for the primary $Cl_2$ isotope ($m/z$ 197, $I^{35}Cl^{35}Cl^-$), the study of $Cl_2$
is limited under these conditions, due to lack of isotopic confirmation of its presence. Though $m/z$ 201 ($I^{37}Cl^{37}Cl^-$) was
additionally monitored, no $Cl_2$ was measured above the limit of detection of $1.1 \pm 0.2$ pmol mol$^{-1}$ for $m/z$ 199 with the
appropriate isotope ratios in experiments with added $O_3$ (6.2 for $m/z$ 199:201).
CIMS calibrations were performed using $I_2$, $Br_2$, and $Cl_2$ permeation devices (VICI) at the start and
conclusion of each experiment. $Br_2$ and $Cl_2$ permeation outputs were quantified using the spectrophotometric method
described by Liao et al. (2012). The $I_2$ permeation output was quantified by flowing the $I_2$ through an impinger
containing a NaHCO$_3$ (30mM)/NaHSO$_3$ (5mM) reducing solution. This solution quantitatively reduces $I_2$ to $I^-$, which
was then determined using a Dionex DX500 ion chromatography system. Permeation rates were calculated for each
experiment and found to average $(1.9\pm0.1) \times 10^{-11}$, $(5.5\pm0.1) \times 10^{-10}$, and $(8.6\pm0.1) \times 10^{-10}$ mol min$^{-1}$ of $I_2$, $Br_2$, and



Cl$_2$, respectively (uncertainties representing standard error of the mean). CIMS calibration factors were calculated for
individual experiments. These factors are based on the average of the signal sensitivities, determined from the
permeation sources, calculated at the start and completion of each experiment. Corresponding uncertainties for these
calibration factors thus represent the 1σ standard deviation of the mean sensitivity. In addition, the sensitivity for
HOBr (as IHOBr$^-$) species is assumed to be a factor of $0.5 \pm 25\%$ the $m/z$ 287 Br$_2$ sensitivity (Liao et al., 2012), though
IHOBr$^-$ was not unambiguously observed according to isotope ratios due to an interference at $m/z$ 223 (IHO$^{81}$Br$^-$). An
approximate I$^{79}$IBr$^-$ calibration factor was assumed to be the average of the sensitivities for $m/z$ 287 (IBr$_2^-$) and 381
(I$_3^-$). Background measurements were performed before and after the experiment (minimum of 5 min) by passing the
carrier gas through the experimental flow tube (without O$_3$, in the dark), and subsequently through a glass wool
scrubber, previously shown to remove molecular halogens with greater than 95% efficiency (Liao et al., 2012; Neuman
et al., 2010). Temporal variations in bromine-species signals while using the low pH sulfate/bisulfate buffer were
observed in some experiments (Fig. S1) and are discussed in the Supplementary Information.

Analysis of experimental data was based on one-minute averages, with uncertainties representing the

standard deviation of these averages. Subsequently, signals were converted to concentrations using the sensitivities
calculated above, propagating the sensitivity uncertainty into the measurement uncertainty. Average limits of
detection (3σ) across all experiments for the molecular halogens during background periods were $1.8 \pm 0.4$, $1.2 \pm 0.3$,
and $9 \pm 2$ pmol mol$^{-1}$ for Br$_2$, Cl$_2$, and I$_2$ respectively (uncertainties representing standard error of the mean).
Additionally, reported uncertainties for integrated amounts of formed halogens are calculated as integrated halogen
concentrations multiplied by the relative uncertainty in the CIMS signal sensitivity.
**3    Results**

The experiments described here address the extent to which OH radicals in the condensed phase can lead to I$_2$,

Br$_2$, and Cl$_2$ production through condensed-phase reactions within frozen saline surfaces, as hypothesized from recent
field (Custard et al., 2017; Pratt et al., 2013; Raso et al., 2017) and laboratory experiments (Abbatt et al., 2010). In
addition, we test the pH-dependence of this chemistry, and whether O$_3$ can enhance this production. We find the
relative and absolute amounts of I$_2$, Br$_2$, and Cl$_2$ produced from ice are a complex function of the relative
concentrations of the precursor halide ions, the pH, presence of oxidants, radiation, and O$_3$.





The ice-coated flow tube experiments started under dark conditions and without addition of $O_3$ (Sect. 3.1). Once
signals stabilized, lights were activated for 1-2 h (Sect. 3.2, Sect. 3.3), after which the ozone generator lamp was
activated to generate ~60 nmol mol$^{-1}$ of $O_3$ in the carrier gas to test the impacts of both radiation and ozone (Sect. 3.4).
Saline surfaces tested include frozen Instant Ocean (IO) solutions, solutions composed of dissolved reagent grade salts
mimicking saltwater (SW) composition, and 0.56 M high purity NaCl (CL1). Unless otherwise specified, integrated
amounts of produced halogens represent amounts produced over the course of 1 h of exposure to light (Sect. 3.2, Sect.
3.3) and/or ozone (Sect. 3.4). Many of the salient features of our results are demonstrated by example experiments
shown in Fig. 2, which shows the impact of irradiation in the presence of ice phase OH radical precursors, varied pH,
and the presence of $O_3$. Below we discuss the details of our experiments, organized by the mechanism of halogen
production and halogen products themselves.

## 235    3.1    Dark reaction production of $I_2$

To photochemically create condensed-phase hydroxyl radicals, either hydrogen peroxide ($H_2O_2$) or nitrite
($NO_2^-$) were added to salt solution samples, as they have been estimated to account for 96% of snowpack
photochemical OH formation at Utqiaġvik, AK (France et al., 2012). However, both $H_2O_2$ and $NO_2^-$ can directly
convert $I^-$ to $I_2$ under dark acidic conditions. No dark production of $Br_2$ or $Cl_2$ was observed in these experiments for
any pH, or presence of OH precursor. The oxidation of $I^-$ by $H_2O_2$ occurs through the condensed phase reactions R23
and R24 (Küpper et al., 1998):
$I^- + H_2O_2 \leftrightarrow HOI + OH^-$     (R23)
$HOI + I^- + H^+ \rightarrow I_2 + H_2O$     (R24)
Nitrite ions react with hydronium ions to form the nitroacidium ion, $H_2ONO^+$, which has been previously shown to
produce $I_2$ (R25-R27) (measured spectrophotometrically as trihalide species, which exist in equilibrium with $X_2$, as in
R12, R17) (Hellebust et al., 2007; O'Driscoll et al., 2006, 2008; O'Sullivan and Sodeau, 2010):
$NO_2^- + H_3O^+ \leftrightarrow HONO + H_2O$     (R25)
$HONO + H_3O^+ \leftrightarrow H_2ONO^+ + H_2O$     (R26)
$2 H_2ONO^+ + 2I^- \leftrightarrow 2NO + I_2 + 2H_2O$     (R27)
The pH $\leq$ 2 experiments in this work favor the forward reactions that produce $I_2$ (R23-24, R25-27).





During the initial connection of the flowtube to the CIMS that would initiate the start of an experiment, a large
$I_2$ signal (measured as $I_3^-$, $m/z$ 381) was observed in several cases in which OH-radical precursors were utilized (e.g.,
Fig. S2, Fig. 2b). This $I_2$ production likely originated from the above reactions (R23-27). This signal subsequently
decayed as $I_2$ flushed out of the system until reaching a steady state. Integrated sums of dark $I_2$ production are
estimated in Table S1. However, these amounts represent lower limits of the true sums of dark-produced $I_2$; while the
flow tube is sealed during the freezing of the salt solutions, this seal is removed during connection to the experimental
flow path (Fig. 1), making it impossible to accurately determine the extent of $I_2$ production prior to irradiation.
At pH ≈ 4.7, dark $I_2$ production was modest, only noticeably affecting experiment IO2 (Fig. S2). In that case,
the integrated sum of $I_2$ released on connection of the flow tube to the CIMS until stabilization was 0.8 (± 0.1) nmol,
corresponding to approximately 0.5% of the total 152 nmol $I^-$ available for reaction from the Instant Ocean solution
(Table S1). At lower pH (<2), larger amounts of $I_2$ were observed in the dark immediately upon flow tube connection
to the CIMS (i.e., before addition of light and $O_3$; Fig. 2b). Dark production of $I_2$ could cause significant depletion of
$I^-$, as Experiments IO4 and SW5 (both using $H_2O_2$) only have, at most, ~46% of the initial 152 nmol of $I^-$ by the time
lights are initiated. These values were calculated by subtracting twice the integrated sums of $I_2$ produced prior to
irradiation (Table S1; i.e., two $I^-$ for every $I_2$) from the total number of $I^-$ moles in the original IO or SW solutions.
Considerably less $I_2$ production occurred in the dark using $NO_2^-$ as an OH precursor (depleting $I^-$ by an average of
4.5%, Table S1). As will be discussed in Sect. 3.3, following depletion of $I^-$ from the salt solutions, $Br_2$ was the
primary halogen produced at low pH.
**3.2   Hydroxyl radical-induced halogen production at pH ≈ 4.7**
Integrated amounts of photochemically produced molecular halogens are presented in Table 2 for all
experiments. Integration times for calculated halogen production span one hour, beginning at the time lights were
activated until 60 minutes later, and in the absence of $O_3$. At pH ≈ 4.7, experiments without hydroxyl radical
precursors (IO6-IO7, SW6-SW7; Table 2) produced amounts of molecular halogens below their respective LODs
from the saline ice surfaces after activation of lights. Experiment IO7 was an exception, however, producing 0.11 ±
0.06 nmol of $I_2$.
In the presence of $H_2O_2$ at pH ≈ 4.7, $I_2$ mole fractions increased rapidly upon irradiation, as shown in Fig. 2a.
Of the four experiments performed in these conditions (IO1, IO2, SW1, SW2), three experiments (IO1, SW1, SW2)





produced statistically similar amounts of $I_2$ (mean: $8 \pm 2$ nmol) after one hour of irradiation (Table 2). Experiment
IO2 (Fig. S2), while experimentally identical to IO1, appears to have produced ~10 times less $I_2$ during this hour after
the lights were turned on. However, $I_2$ was already present prior to turning on the lights, suggesting production
originating from the direct reaction between $I^-$ and $H_2O_2$ (Sect. 3.1). Experiment IO2 otherwise eventually qualitatively
resembles the other three analogous experiments (IO1, SW1, SW2; e.g. Fig. 2a) with the $I_2$ concentration eventually
increasing after irradiation (Fig. S2).

Regarding other molecular halogens, IBr was observed upon radiation during Experiment SW2 (Fig. 2a)

above the estimated limits of detection (3 pmol mol$^{-1}$) starting approximately 20 minutes before the addition of $O_3$.
No direct (OH-induced) photochemically produced $Br_2$ was unambiguously observed at this pH. The apparent
photochemical integrated $Br_2$ sum of $0.034 \pm 0.003$ nmol reported for experiment IO2 (Table 2) stems from a real
signal just above the limit of detection ($1.8 \pm 0.4$ pmol mol$^{-1}$), and this baseline signal does not change on addition of
light. This signal, however, remains below limits of quantitation and should not be considered further. $Cl_2$ mole
fractions remained below limits of detection in all cases with OH-precursors at this pH.
**3.3    Effects of the hydroxyl radical on halogen production at pH < 2**

It is expected that halogen production will be enhanced when pH is decreased based on reactions R4-R22. In

cases without OH precursors at pH < 2, photochemical $I_2$ production was observed (integrated production of $14 \pm 10$
nmol for IO8, and $6.0 \pm 2.0$ nmol for SW8) (Table 2), in contrast to experiments performed at pH = 4.7 in which very
little was produced. This production likely stems from the mechanisms outlined by Kim et al. (2016) (R13-17), which
requires only light and oxygen to form a charge-transfer complex that results in $I_2$ production (discussed in Sect. 1).
Molecular $Br_2$ and $Cl_2$ concentrations remain below limits of detection, consistent with Abbatt et al. (2010), in which
no $Br_2$ or $Cl_2$ was observed without an OH-precursor.

As discussed in Sect. 3.1, inclusion of $H_2O_2$ or $NO_2^-$ can result in direct oxidation of $I^-$, thereby reducing the

available $[I^-]$ for photochemical OH oxidation when pH < 2. When $H_2O_2$ was used as an OH precursor, photochemical
production of $I_2$ across experiments yielded $\leq 0.82$ nmol (IO4, IO5, and SW5), likely due to the dark $I_2$ production
mechanisms. When instead $NO_2^-$ was used (as in IO3 and SW3, SW4), initial observations of $I_2$ on flowtube
connection to CIMS were as much as 90% less than when $H_2O_2$ was used (Table S1). For experiment IO3, the reduced
pH enhanced $I_2$ production ($39 \pm 1$ nmol) compared to the high pH cases (Experiments IO1-2, SW1-2, ranging from



0.6 ± 0.4 to 9 ± 3 nmol) (Table 2). The corresponding seawater experiments were not as conclusive; experiment SW3
only yielded 4.0 ± 0.1 nmol of photochemical $I_2$ (Fig. S4). Experiment SW4 did not produce any photochemical $I_2$
and qualitatively resembles the low pH $H_2O_2$ cases. It is possible that for SW3 and SW4 more $I_2$ was produced by
dark reactions and flushed out of the tube during connection with the CIMS and was therefore not measured.

When [$I^-$]/[$Br^-$] approximates the initial conditions of Instant Ocean (~2.6 x $10^{-3}$; i.e., IO1-IO3, SW1-SW3),

OH-mediated $I_2$ production precedes $Br_2$ and IBr production. This initial photoproduction of $I_2$ is observed for IO4,
as shown in the inset of Fig. 2b. Figure 2b shows a delay in $Br_2$ production until $I^-$ was removed as $I_2$ (and IBr). After
[$I^-$]/[$Br^-$] has sufficiently decreased, $Br_2$ eventually becomes the dominant photochemical product, yielding an average
of 4.5 ± 0.5 nmol from IO4 and IO5, and 6.0 ± 0.7 nmol from SW5, all of which used $H_2O_2$ as an OH precursor, and
5.4 ± 0.7 nmol from SW4, which used $NO_2^-$ as an OH precursor. Simultaneous production of IBr was observed as
well (Fig. 2b) when [$I^-$]/[$Br^-$] had been reduced following $I_2$ production. Given the initial depletion of $I^-$ from dark $I_2$
production (Sect. 3.1), we can estimate [$I^-$]/[$Br^-$] at pH < 2 in ice with $H_2O_2$ just before irradiation based on the
remaining moles of $I^-$ in solution (Table S1) and the initial moles of $Br^-$ (calculated using the initial experiment solution
volume and concentration of $Br^-$ of 7.2 x $10^{-4}$ M). [$I^-$]/[$Br^-$] was calculated as (1.6 ± 0.7) x $10^{-4}$, which is an average
calculated for experiments IO4-5 and SW5, and was sufficiently low to result in photochemical production of $Br_2$.

Photochemical $Cl_2$ production was only observed from a frozen solution of "pure" NaCl and $H_2O_2$ at pH=1.8

(CL1), as shown in Fig. 2c. When the lights were turned on, a slight increase in $I_2$ and IBr were observed in concert
with a rapid rise in $Br_2$, likely resulting from an $I^-$ impurity in the NaCl salt. After about one hour of apparent
equilibrium, $I_2$ concentrations begin decreasing, while $Br_2$, IBr, and $Cl_2$ continue rising. Over one hour of illumination,
93 ± 3 pmol of $Cl_2$, 100 ± 10 pmol of $Br_2$, and 100 ± 10 pmol of $I_2$ were measured. However, as shown in Fig 2c, the
greatest rate of increase in signal did not occur until t = 1 h (after irradiation). Integrating instead from t=0 until t=2
hours, the amount of $Cl_2$ produced was 190 ± 10 pmol, while the amount of $Br_2$ increased to 310 ± 20 pmol. The
initial [$Br^-$] of the CL1 solution was determined to be (4.5 ± 0.3) x $10^{-6}$ M via ion chromatography, meaning $Cl_2$
production was observed at [$Br^-$]/[$Cl^-$] of 8.1 x $10^{-6}$ ($^1/_{124,000}$), compared to the Instant Ocean [$Br^-$]/[$Cl^-$] of ~ $^1/_{800}$.
Unfortunately, BrCl could not be observed due to an unknown interference at *m/z* 241 and 243.



### 3.4 Effects of $O_3$ on halogen production


In experiments without an OH source (IO6-IO8, SW6-SW8), $I_2$ production was greatest when $O_3$ was
introduced to the irradiated tube for both pH regimes (Table 2). The amount of $I_2$ produced in these experiments was
large, ranging from $26 \pm 9$ nmol to $80 \pm 1$ nmol at pH $= 4.7$, and from $2.6 \pm 1.7$ nmol to $38 \pm 12$ nmol at pH $< 2$. While
the $I_2$ produced pH $< 2$ appears to be lower, $I_2$ had already been produced prior to addition of $O_3$ (i.e., with only light
as a stimulant, Sect. 3.3), yielding a lower $[I^-]/[Br^-]$ ratio when $O_3$ was eventually added. $Br_2$ production amounts
ranged from $0.012 \pm 0.001$ nmol to $0.16 \pm 0.01$ nmol at pH $= 4.7$ and taking up to 6 hours to raise above detection
limits after $O_3$ was added. At pH $\leq 2$, $Br_2$ production amounts ranged $0.14 \pm 0.02$ nmol to $0.93 \pm 0.05$ nmol, in the
absence of an OH source.
When OH-precursors were present, the addition of $O_3$ to the zero-air flow over the irradiated frozen sample
caused additional production of $I_2$ and $Br_2$, as shown in Figure 2a and b, under both pH conditions (pH $\leq 2$, pH $= 4.7$)
(Table 2). $I_2$ integration times here represent one hour, beginning at the time when $O_3$ is introduced until 60 minutes
later. In experiments at pH $\approx 4.7$ in which $[I^-]/[Br^-]$ remained sufficiently large due to minimal dark production of $I_2$
(i.e., IO1-2, SW1-2), exposure to $O_3$ caused a sharp increase in $I_2$ (as in Fig. 2a). $I_2$ production amounts for frozen
Instant Ocean at pH $\approx 4.7$ (IO1, IO2) average $22 \pm 10$ nmol, about two times less than for frozen saltwater experiments
SW1 and SW2 (average production amount of $51 \pm 25$ nmol). As the $I_2$ signal decayed, the corresponding $Br_2$ signals
gradually increased above detection limits, approximately 3h after the introduction of $O_3$ (Fig. 2a). The average
integrated amounts of $Br_2$ produced from these pH $\approx 4.7$ experiments were very similar ($0.05 \pm 0.01$ nmol for IO
experiments and $0.03 \pm 0.01$ nmol for SW experiments).
When pH $< 2$, the effects of $O_3$ addition varied according to the remaining availability of $I^-$. When the surface
$I^-$ reservoir had been reduced from dark reactions with $H_2O_2$ or $NO_2^-$ (R17-21; Sect. 3.1), exposure to $O_3$ did not
increase $I_2$ above the LOD in all experiments except IO5, which exhibited a small spike before decaying below the
LOD ($0.11 \pm 0.06$ nmol in IO5). However, $O_3$ did cause additional $Br_2$ production after one hour (average of $10 \pm 2$
nmol for IO4 (Fig. 2b) and IO5, and $14 \pm 2$ nmol for SW4 and SW5). In contrast, for SW3 (using $NO_2^-$ as an OH
source), there was relatively little initial consumption of $I^-$ by dark reaction; therefore, when $O_3$ was added, an amount
of $I_2$ equal to $1.1 \pm 0.1$ nmol was observed, comparable to what was observed with the higher pH experiments (Fig.
S4). The amount of $Br_2$ produced ($0.46 \pm 0.01$ nmol) was also significantly less than observed when $I^-$ was initially
depleted, demonstrating the importance of the halide ratios (see Section 4.2). Unfortunately, the addition of $O_3$



introduced a strong interference for the signal observed at $m/z$ 197 ($I^{35}Cl^{35}Cl^-$) rendering $Cl_2$ isotopic ratios invalid,
and hence no information regarding the relationship between $Cl_2$ and $O_3$ could be ascertained for any experiments
involving $O_3$.
HOX compounds were also observed when $O_3$ was present, likely formed in the flowtube by $O_3$ reactions
with halides as in R18 and R19 (Fig. 3-4; discussed in more detail in the Supplemental Information).  Figure 3 shows
this for IO2 (pH=4.7 experiment using Instant Ocean, analogous to SW2 in Fig. 2a, as well as IO1 and SW1).  For
each experiment in this series (pH=4.7 with OH-precursors), increases in $I_2$, HOI, and $Br_2$ were readily observed when
the $O_3$ was introduced at hour 2 (Fig. 3).  However, corresponding HOBr production was not observed, perhaps either
due to a high LOD, or the relatively low abundance of $Br_2$ that would limit production of HOBr.  Conversely, in pH
≤ 2 cases when substantial portions of $I^-$ had already reacted prior to irradiation (IO4, IO5, SW4, SW5), the addition
of $O_3$ produced negligible amounts of $I_2$ and HOI (Fig. 4).  But, in these cases, following the addition of $O_3$, HOBr
($m/z$ 225 IHO$^{81}$Br$^-$), was observed together with $Br_2$ (Fig. 4).  We note in this case that $m/z$ 223, representative of
IHO$^{79}$Br$^-$, does not appear to show an enhancement when $O_3$ is added to the system.  There was a much higher
background signal for $m/z$ 223 compared with $m/z$ 225 (IHO$^{81}$Br$^-$) resulting from an unknown interference.
**4    Discussion**
**4.1    Role of OH in halogen production in ice**
The observations in this study indicate competition for the OH radical in which the most oxidizable halide is
oxidized, and the corresponding molecular halogens are produced until that halide is depleted at the surface.  The
trends in molecular halogen production show acid-enhanced production mechanisms, in which the dominant products
are largely dependent on relative halide ratios.  These results are consistent with in situ observations of $Br_2$, BrCl, and
$Cl_2$ formation (Custard et al., 2017; Pratt et al., 2013).  In the case of this work, $Br_2$ and IBr were not observed until $I_2$
production sufficiently decreased the $[I^-]/[Br^-]$ ratio, and $Cl_2$ was not observed unless the $[Br^-]/[Cl^-]$ ratio was
sufficiently low ($[Br^-]/[Cl^-]$ = 8.1 x $10^{-6}$ in this study).  This observation is consistent with other lab studies (Abbatt et
al., 2010; Sjostedt and Abbatt, 2008).  Sjostedt and Abbatt (2008) exposed frozen salt solutions to gas-phase OH and
found peak BrCl production occurred as $Br^-$ decreased from an initial $[Br^-]/[Cl^-]$ of 7.3 x $10^{-5}$.  Abbatt et al. (2010)
generated condensed phase OH on frozen surfaces from photolysis of nitrate, and similarly found lower $Br_2$ and IBr



integrated amounts at lower [Br⁻]/[Cl⁻] when temperatures were warmer than the eutectic point of sodium chloride.
However, the I⁻ in Abbatt et al. (2010) originated from impurities in their sodium chloride and sodium bromide
reagents and was not quantified, making the relative ratios regarding I⁻ not quantifiable.

As a first approximation, we estimate via Eq. (1) effective relative reaction rate constants ($k_{X^-}/k_{Y^-}$, where X

and Y represent Br, Cl, or I) for reaction of a halide with OH radicals, assuming that the observed $X_2$ flux out of the
ice is proportional to the production rate (i.e., $X_2$ desorbs as it is produced, within the residence time of the flow tube),
and that oxidation by OH is the rate limiting step:
$$\frac{Flux_{X_2}}{Flux_{Y_2}} = \frac{k_{X^-}[X^-][OH][H^+]}{k_{Y^-}[Y^-][OH][H^+]}$$                    (1)
The assumption that OH oxidation is rate limiting is based on individual $I_2$ and $Br_2$ photochemical production amounts
between Instant Ocean and saltwater solutions not being statistically different (i.e., the organic matter in Instant Ocean
does not appear to impact halogen production), and its dependence on radiation and presence of an OH precursor. The
initial molecular halogen flux is calculated as the integrated sum of $X_2$ (in moles) divided by both integration time (t
= 0-3 minutes, starting from the beginning of irradiation to capture the initial flux) and the surface area of ice coverage
in the flow tube. The surface area, as well as the [OH] and [H⁺] in the "disordered interface" would be identical within
individual experiments and cancel in these calculations. The halide ion concentrations (defined in Sect. 2) allow us
to solve for the effective relative rate constant, $k_{X^-}/k_{Y^-}$ by assuming the ratios of the halide concentrations are the same
as in the pre-freezing solution. At pH = 1.8, we estimate $k_{Br^-}/k_{Cl^-}$ = (2.4 ± 0.2) x $10^5$ from experiment CL1, or, in other
words, production of $Br_2$ is 240,000 times more efficient than production of $Cl_2$ via (OH + halide) in the surface layer.
Across the six experiments performed at pH < 2 (average of 1.85) using Instant Ocean (IO3, IO4, IO5) and saltwater
(SW3, SW4, SW5), we calculate an average $k_{I^-}/k_{Br^-}$ of (9 ± 4) x $10^3$ (reported uncertainty is the standard error of the
mean, and thus only represents the experiment repeatability).

The above relative rate constant calculations are considered upper limits since the halide ratios used represent

those in the pre-freezing solution. In other words, it is assumed that the ions are excluded to the disordered interface
in amounts proportional to their pre-freezing concentration. Malley et al. (2018) recently demonstrated that brine can
be distributed throughout ice in channels, suggesting that only the solutes at the liquid-air interface (a fraction of the
total pre-freezing solution) participate in heterogeneous chemistry. Indeed, we find evidence here suggesting not all
ions are available for reaction at the disordered interface surface, particularly experiments for which we lost little I⁻



from dark I$_2$ production mechanisms (i.e., pH = 4.7 with OH precursors: IO1, IO2, SW1, SW2). Considering
experiment IO2 as an example (Fig. S5), integration of the I$_2$ signal during ~15 hours of exposure to both light and O$_3$
shows that 54% (82 nmol) of the original 152 nmol of I$^-$ remained unreacted in the frozen solution despite the signal
apparently stabilizing at its baseline. A similar calculation cannot be performed for the pH ≤ 2 experiments because
of the inability to accurately quantify the amount of I$_2$ lost during connection of the flowtube to the CIMS. It is
therefore probable that a significant number of the ions, as well as H$_2$O$_2$, exist within brine channels within the ice
(Bartels-Rausch et al., 2014; Malley et al., 2018), such that oxidation chemistry is occurring throughout the ice. The
diffusion rates of the product molecular halogens through bulk ice are likely slow, such that only surface production
is observed here (Abbatt et al., 2012). Henry's Law constants suggest that I$_2$ not at the ice surface will transfer to the
air more slowly than other molecular halogens due to having a higher solubility (41.9 M/atm and 8.4 M/atm at -20 ºC
for I$_2$ and Br$_2$. respectively) (Raso et al., 2017). Consequently, upon irradiation, OH radicals will react with the most
oxidizable ion via R8-12 (e.g., I$^-$). Of the halogens produced from frozen solutions here, it is expected that I$_2$ is
observed most readily given the high polarizability and surface affinity of I$^-$ in aqueous solutions (Gladich et al., 2011),
and the relative ease of oxidation of I$^-$. That is, surface concentrations will be relatively enhanced with larger, more
polarizable anions (I$^-$ > Br$^-$ > Cl$^-$) (Gladich et al., 2011), which favors production of I$_2$ over Br$_2$, and Br$_2$ over Cl$_2$. As
the larger/more reactive ions are depleted through oxidation, the next largest ion then becomes more favorably
oxidized. This implies that, if the larger anion is enhanced at the surface, the calculated relative rate constants do not
accurately represent fundamental relative reactivity, but rather the effective relative reactivity given knowledge of the
bulk composition. However, the observed relative oxidation rate constants are consistent with the standard reduction
potentials (tendency to become oxidized) for I$^-$, Br$^-$, and Cl$^-$, i.e. 0.620, 1.098, and 1.360V for I$_2$, Br$_2$, and Cl$_2$,
respectively (Chemical Rubber Company and Lide, 2005).

Despite these relative oxidation rates, molecular halogen levels have been previously observed at

concentrations within the snowpack interstitial air within two orders of magnitude of each other. At Utqiaġvik, AK,
snowpack Br$_2$ has been observed under artificial radiation at peak levels of 1100 pmol mol$^{-1}$ (Custard et al., 2017), I$_2$
up to 50 pmol mol$^{-1}$ (Raso et al., 2017) , and Cl$_2$ up to 20 pmol mol$^{-1}$ (Custard et al., 2017) under artificial irradiation.
Though there are substantially lower natural abundances of I$^-$ (Raso et al., 2017), I$_2$ is still observed at levels
comparable to / approaching those typical of Cl$_2$ and Br$_2$. The relative rate constants (ease of X$^-$ oxidation by OH
radicals) we calculate would then appear to explain that the reactivity of the larger ions (which incorporates



components of surface affinity and chemical reactivity to OH) compensate for the low abundances, possibly leading
to comparable production rates in our laboratory experiments, and comparable snowpack gas phase concentrations.

Using a modified version of Eq. (1), we can estimate relative in situ OH-mediated halogen production rates

using published halide ratios from melted in situ snow samples from Utqiaġvik, AK (Eq. (2)).
$$\frac{\frac{d[X_2]}{dt}}{\frac{d[Y_2]}{dt}} = \frac{k_{X^-}[X^-][OH][H^+]}{k_{Y^-}[Y^-][OH][H^+]}$$    (2)
For a range of possible values, we utilize previously published, minimum and maximum $[Br^-]/[Cl^-]$ values that include
corresponding $X_2$ observations (i.e., halide ratios from samples that were shown to photochemically produce $X_2$):
$0.0005 \pm 0.0001$ (Custard et al., 2017) and $0.026 \pm 0.008$ (Pratt et al., 2013). These ratios yield a corresponding range
for $\frac{\frac{d[Br_2]}{dt}}{\frac{d[Cl_2]}{dt}}$ of $180 \pm 20$ to $6000 \pm 2000$. $\frac{\frac{d[I_2]}{dt}}{\frac{d[Br_2]}{dt}}$ can additionally be estimated from Raso et al. (2017), where $[I^-]/[Br^-]$
ranges from $0.00040 \pm 0.00003$ to $0.129 \pm 0.006$, calculated $\frac{\frac{d[I_2]}{dt}}{\frac{d[Br_2]}{dt}}$ values range from $3.3 \pm 1.5$ to $1200 \pm 500$. While
simultaneous production of $I_2$ and $Cl_2$ was not observed herein, $k_{I^-}/k_{Cl^-}$ can be calculated by multiplying $k_{I^-}/k_{Br^-}$ by $k_{Br^-}$
$/k_{Cl^-}$. Therefore, for $[I^-]/[Cl^-]$ ranging from $(1.0 \pm 0.1) \times 10^{-6}$ to $(1.5 \pm 0.1) \times 10^{-4}$ (Raso et al., 2017), we obtain $\frac{\frac{d[I_2]}{dt}}{\frac{d[Cl_2]}{dt}}$
values of $(2 \pm 1) \times 10^3$ to $(3 \pm 2) \times 10^5$. The results of these calculations indicate that the observed relative rates of
production more than compensate for the relative halide ion abundances.

However, the observed relative rates of production are inconsistent with the observed relative snowpack

interstitial air $X_2$ abundances from field observations, which show similar (within a factor of 10) abundances in
irradiated snowpack interstitial air. We can formulate the following hypotheses to explain this:

1. There exist important competing loss processes for $Br_2$ and $I_2$ after initial production.

2. Other $Cl_2$ production pathways account for the majority of ambient concentrations.

Concerning hypothesis 1, one likely loss process includes aqueous inter-halogen partitioning, as in R25 (where X = I
or Br, and Y = I, Br, or Cl):
$X_2 + Y^- \leftrightarrow X_2Y^- \leftrightarrow XY + X^-$    (R28)
Evidence supporting this possibility include the photochemical formation of IBr in concert with photochemical $Br_2$
production during low pH experiments (in Fig. 2b), as has been observed previously (Abbatt et al., 2010; Sjostedt and
Abbatt, 2008). In addition, photolysis of $X_2$ is faster for the larger molecular halogens. Thompson et al., (2015) report





$J_{X_2}$ = 0.15, 0.044, and 0.0021 s$^{-1}$ for I$_2$, Br$_2$, and Cl$_2$, respectively during solar noon in March at Utqiaġvik, AK,
corresponding to photolytic lifetimes of 7s, 23s, and 476s, respectively.  Thus, faster photolysis of the larger halogens
in the snowpack air will contribute to levelling the production rate differences, given penetration of actinic radiation
into the snowpack (King and Simpson, 2001).  Evidence also exists in support hypothesis 2, possibly via the
heterogeneous recycling involved in the halogen explosion mechanism outlined in Sect. 1. Wang and Pratt (2017)
discuss that Br$_2$ and Cl$_2$ in ambient air at Utqiaġvik have opposite diurnal trends, indicating different governing
mechanisms for each molecular halogen species. We discuss this in greater detail in Sect. 4.2.
**4.2     The role of O$_3$ in enhancing halogen production**
In experiments without an OH source, I$_2$ production amounts were greatest after O$_3$ was introduced to the
illuminated tube for both pH regimes (Table 2).  This likely results from a combination of heterogeneous recycling,
and the surface and aqueous reactions between O$_3$ and I$^-$  (k = 2.0 x 10$^{-12}$ cm$^3$ molecules$^{-1}$ s$^{-1}$ (Liu et al., 2001)).  While
O$_3$-mediated halogen production has been observed directly from frozen surfaces in previous laboratory studies
(Artiglia et al., 2017; Oldridge and Abbatt, 2011; Oum et al., 1998a; Wren et al., 2013), Br$_2$ was not observed to be
produced from the Arctic snowpack without irradiation (Pratt et al., 2013).  This discrepancy raises a question of the
role of O$_3$ in initial halogen release in the Arctic spring.
In the presence of light, O$_3$ was found to stimulate additional I$_2$ and Br$_2$ production in the experiments herein,
as discussed in Sect 3.4.  This additional production could result from a combination of the following mechanisms.
First, as discussed above, O$_3$ can react with halides on frozen saline surfaces to produce Br$_2$ or I$_2$ per reactions R18-
19, and then R4 (Artiglia et al., 2017; Carpenter et al., 2013; Gladich et al., 2015; Hayase et al., 2010; Oum et al.,
1998a; Shaw and Carpenter, 2013; Wren et al., 2013).  It is possible that Br$_2$ (as well as other halogens) may have
been produced via this mechanism at levels below the LOD in previous Arctic snowpack studies (Custard et al., 2017;
Pratt et al., 2013; Raso et al., 2017); however, this may provide sufficient levels of Br$_2$ to enrich the snowpack in Br$^-$
to drive the photochemical production mechanism upon radiation.  Second, given a flow tube residence time of 12
seconds, gas phase production of HOX is possible and could potentially enhance X$_2$ production, given a timescale for
molecular diffusion of 6.5 seconds for HOBr from the center of the tube to the ice surface.  At this flow rate, there is
enough time for 1-2 heterogeneous reaction cycles. Consistent with this recycling mechanism, we observed HOI, and
HOBr at low pH (Fig. 3-4).



Revisiting the pathways for $Cl_2$ production from Sect. 4.1, there is growing evidence for the role of heterogeneous
chemistry. Recently, Wren et al. (2013) observed substantial $Cl_2$ production from their artificial snow samples in the
presence of both $O_3$ and light, invoking the "halogen explosion" mechanism (Sect. 1).  In this scenario, HOI or HOBr
could liberate Cl from the "disordered interface" to produce ICl or BrCl, which can undergo R1-4 to produce HOCl
that ultimately oxidizes $Cl^-$ to produce $Cl_2$.  Liao et al. (2014) observed results consistent with this mechanism above
the snowpack in Utqiaġvik, AK, reporting a strong correlation between $Cl_2$, $O_3$, and solar radiation; similarly, Custard
et al. (2016) observed ClO correlated with $Cl_2$.  This pathway is certainly viable for producing $Cl_2$ in our experiments,
given there is enough time for recycling.
## 5    Summary and Conclusions
We show here that the hydroxyl radical can act as an effective condensed-phase halide oxidant leading to $I_2$,
IBr, $Br_2$, and $Cl_2$ production under acidic conditions.  Rates of release were dictated by both pH and relative halide
concentrations.  The molecular halogen produced appears to be highly influenced by which ions are enhanced at the
ice surface, with $I_2$ production occurring prior to $Br_2$ production, which commenced after the $[I^-]/[Br^-]$ was reduced.
An opportunity exists to further explore this chemistry via surface-sensitive methods, for which recent developments
have been shown to effectively enable characterization of the surface composition of frozen solutions of sodium
chloride under near atmospherically relevant conditions (Artiglia et al., 2017; Orlando et al., 2016).  It would be useful
to confirm the dominant ions involved in this surface-based chemistry over time.  Further investigations into the effects
of halide ratios on halogen production are also suggested, including measurements of how the ratios vary for different
frozen Arctic surfaces, as well as how they vary spatially.  While condensed-phase OH produces $Br_2$ and $I_2$ most
rapidly in this study, it appears that other mechanisms, such as heterogeneous recycling of HOCl or $ClONO_2$, could
be a more dominant mechanism for in situ production of gas phase $Cl_2$ (Wang and Pratt, 2017). We find the addition
of $O_3$ provides additional production of at least $Br_2$ and $I_2$, probably through gas-phase production of HOX or $XONO_2$
and subsequent halogen explosion chemistry.  These results lend support for the photochemical mechanisms proposed
by the recent in situ snowpack experiments (Custard et al., 2017; Pratt et al., 2013; Raso et al., 2017) in which
condensed-phase OH chemistry provides seed halogens that subsequently undergo heterogeneous recycling in order
to build up atmospheric concentrations of halogens.



The pH dependence of halogen activation necessitates study on the pH on relevant Arctic frozen surfaces.
Pratt et al. (2013) found that the frozen surfaces most conducive to in situ photochemical $Br_2$ production had acidic
pH after melting, while no production was observed from those with a well-buffered alkaline ice brine.  Similarly, we
find herein that condensed-phase OH-induced halogen production is enhanced at lower pH.  Wren and Donaldson
(2012a, 2012b) found in laboratory studies that pH of acidic and basic solutions remains essentially unchanged after
freezing, and that saline solutions with buffers (i.e., seawater) maintain their buffering capacity following trace gas
deposition, supporting the lack of observed $Br_2$ production from the sea ice surface (Pratt et al., 2013).  Therefore, it
would be useful to test in situ production of halogens from Arctic frozen surfaces in tandem with the testing of the pH
of said surfaces in order to determine the atmospherically relevant surface pH range required for halogen production.

*Data availability*.  The data analysed in this work have been submitted for deposit onto the National Science
Foundation Arctic Data Center (arcticdata.io) for public accessibility.  Until they are published, data are available
upon e-mail request to the first author (halfacre@ius.edu).

*Author contributions*.  JWH and PBS designed the research and JWH performed the experiments and data
analysis.  All authors contributed to the discussion and interpretation of the results and writing of the paper.

*Competing interests*. The authors declare that they have no conflict of interest.
**Acknowledgements**
We would like to thank the National Science Foundation for their funding (PLR-1417668 and PLR-1417906,
OPP-1417668).  We also express thanks to J. H. Slade, L. G. Huey, D. J. Tanner, F. Xiong, A. R. W. Raso, and K. D.
Custard for their assistance with CIMS operation and maintenance. Additionally, we thank the Purdue Chemistry Shop
for helping build both the cooling and photolysis boxes, as well as the Jonathan Amy Facility for Chemical
Instrumentation for their support in the fabrication of the experimental flow tube and setup of our experimental boxes.
Thanks are also extended to M. Haas and M. Bischoff for performing total organic carbon analysis of our samples,

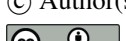



and A. R. W. Raso for confirmation of the iodide concentrations in our Instant Ocean samples.  Finally, we thank T.
Miller and the Purdue Birck Nanotechnology Center for the provision of the nano-grade water used for our samples.

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





**Tables**
Table 1: List of relevant species monitored by chemical ionization mass spectrometry ($I(H_2O)_n^-$ as reagent ion) with corresponding
*m/z* values.

| Species | *m/z* |
|---|---|
| $I^{81}Br^-$ | 208 |
| $I^{79}Br^{79}Br^-$ | 285 |
| $I^{79}Br^{81}Br^-$ | 287 |
| $I^{35}Cl^-$ | 162 |
| $I^{37}Cl^-$ | 164 |
| $I^{35}Cl^{35}Cl^-$ | 197 |
| $I^{35}Cl^{37}Cl^-$ | 199 |
| $I^{37}Cl^{37}Cl^-$ | 201 |
| $I^{79}Br^{35}Cl^-$ | 241 |
| $I^{81}Br^{35}Cl^- / I^{79}Br^{37}Cl^-$ | 243 |
| $I_3^-$ | 381 |
| $IHO^{79}Br$ | 223 |
| $IHO^{81}Br$ | 225 |
| $IHO_3^5Cl^-$ | 179 |
| $IHO_3^7Cl^-$ | 181 |
| $IHOI^-$ | 271 |
| $I^{79}IBr^-$ | 333 |
| $I^{81}IBr^-$ | 335 |







Table 2: Results for all experiments performed. The first line in an experiment represents the integrated totals of molecular halogen
production after 1 hour of irradiation (t = 0 through t =1 h).  The results on italicized lines are 1 h integrated production amounts
beginning once additional ozone was introduced to the flow tube.  Average LODs across experiments were 1.8 ± 0.4, 1.2 ± 0.3,
and 9 ± 2 pmol mol$^{-1}$ for Br$_2$, Cl$_2$, and I$_2$ respectively. "IO#" represents samples composed of Instant Ocean, and "SW#" represents
"saltwater" samples, composed of reagent salts.  "CL1" here represents the experiment performed using 0.557M high purity NaCl.

| Experiment | Oxidant | pH | I$_2$ produced (nmol) | Br$_2$ produced (nmol) | Cl$_2$ produced (nmol) |
|---|---|---|---|---|---|
| IO1 | H$_2$O$_2$ +O$_3$ | 4.7 | 9 (±3) *22 (±8)* | < LOD *0.06 (±0.05)* | |
| IO2 | H$_2$O$_2$ +O$_3$ | 4.7 | 0.6 (±0.4) *21 (±14)* | 0.034 (±0.003) *0.038 (±0.003)* | |
| SW1 | H$_2$O$_2$ +O$_3$ | 4.7 | 6.0 (±2.1) *51 (±19)* | < LOD *0.024 (±0.014)* | |
| SW2 | H$_2$O$_2$ +O$_3$ | 4.5 | 8 (±4) *51 (±25)* | < LOD *0.018 (±0.003)* | |
| IO3 | NO$_2^-$ | 2.0 | 39 (±1) | 0.084 (±0.002) | |
| IO4 | H$_2$O$_2$ +O$_3$ | 1.7 | 0.8 (±0.3) *< LOD* | 5.6 (±0.3) *12 (±1)* | |
| IO5 | H$_2$O$_2$ +O$_3$ | 1.7 | 0.33 (±0.11) *0.11 (±0.04)* | 3.5 (±0.4) *9.2 (±1.0)* | |
| SW3 | NO$_2^-$ +O$_3$ | 1.8 | 4.0 (±0.1) *< LOD* | < LOD *0.46 (±0.1)* | |
| SW4 | NO$_2^-$ +O$_3$ | 2.2 | < LOD *< LOD* | 5.4 (±0.7) *13 (±2)* | |
| SW5 | H$_2$O$_2$ +O$_3$ | 1.8 | 0.75 (±0.26) *< LOD* | 6.0 (±0.7) *15 (±2)* | |
| CL1 | H$_2$O$_2$ | 1.8 | 0.10 (±0.03) | 0.10 (±0.01) | 0.093 (±0.008) |
| IO6 | None +O$_3$ | 4.7 | < LOD *26 (±9)* | < LOD *0.015 (±0.001)* | |
| IO7 | None +O$_3$ | 4.7 | 0.10 (±0.06) *47 (±29)* | < LOD *0.012 (±0.001)* | |
| SW6 | None +O$_3$ | 4.7 | < LOD *80 (±1)* | < LOD *0.16 (±0.01)* | |
| SW7 | None +O$_3$ | 4.5 | < LOD *48 (±2)* | < LOD *0.023 (±0.001)* | |
| IO8 | None +O$_3$ | 2.0 | 14 (±10) *2.6 (±1.7)* | < LOD *0.14 (±0.02)* | |
| SW8 | None +O$_3$ | 2.0 | 14 (±10) *2.6 (±1.7)* | < LOD *0.14 (±0.02)* | |






**Figures**

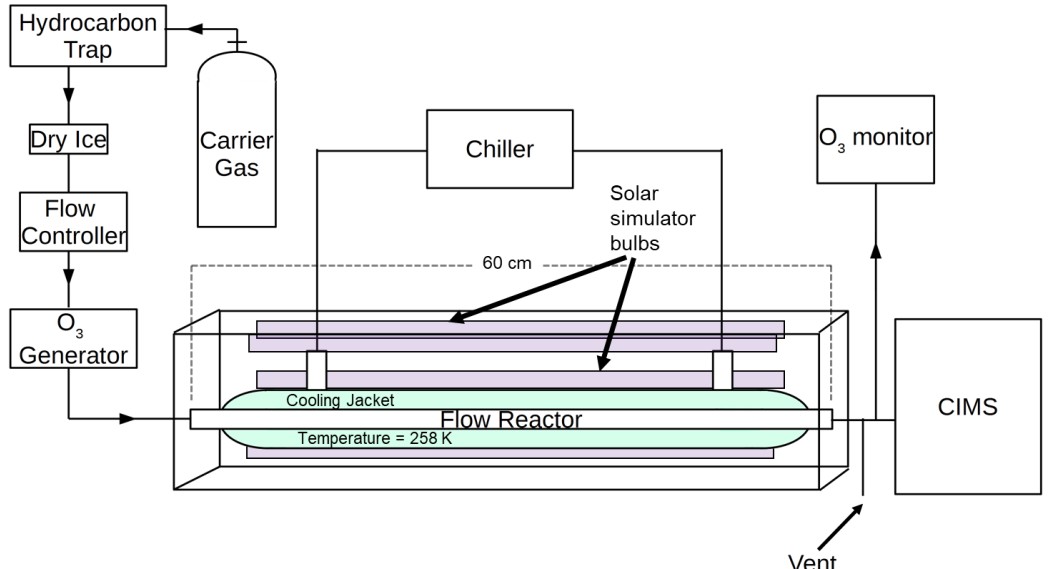


Figure 1: Experimental schematic.  Purple bars represent powered solar simulator bulbs.  The green shading around the flow tube
(flow reactor) represents cooling liquid (60% ethylene glycol, 40% water) circulated through the chiller.  The flow reactor region
itself has an inner diameter of 2.5 cm.



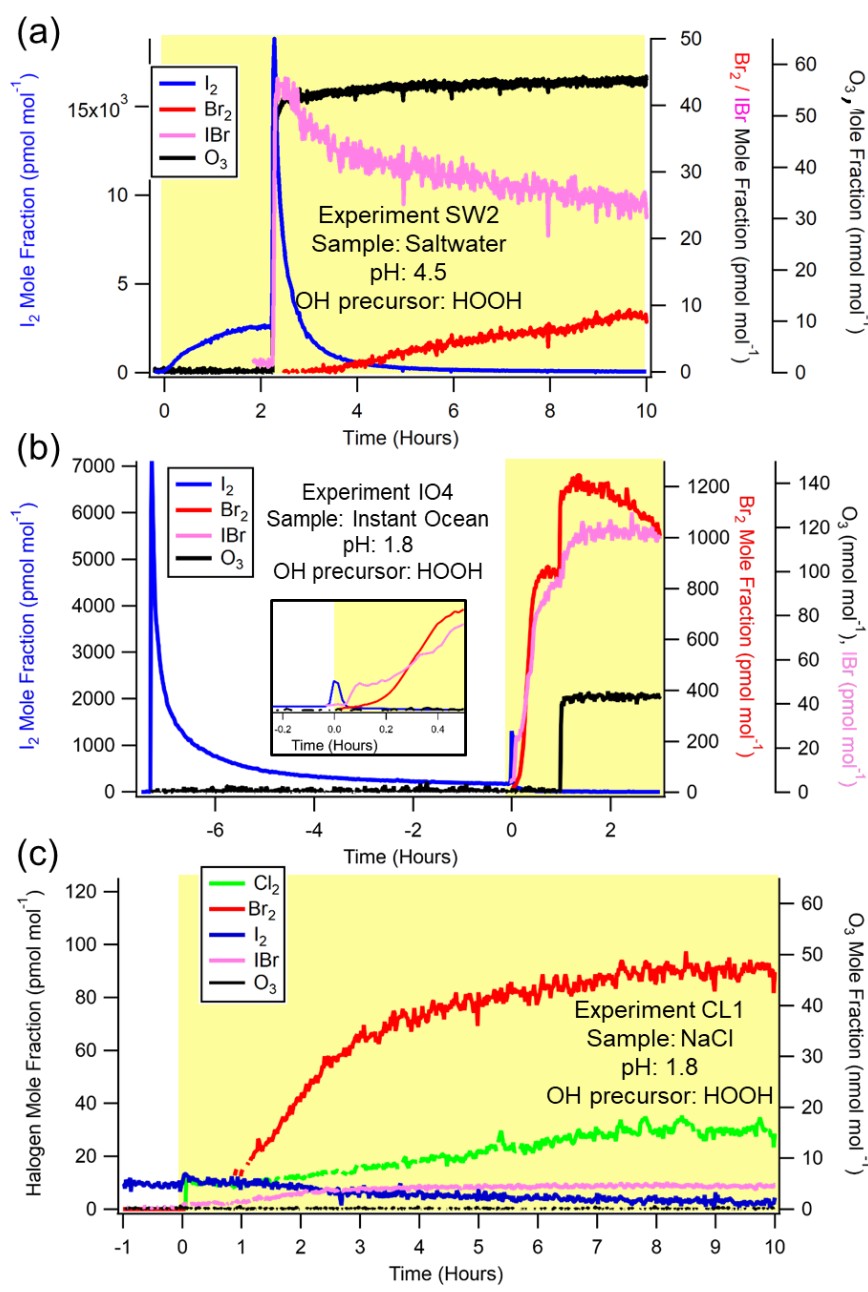

Figure 2: Representative experiments of OH-mediated production of X₂, and subsequent production of X₂ from O₃ addition.  a)
Saltwater experiment (SW2) at pH=4.5.  b) Instant Ocean experiment (IO4) at pH = 1.8. Time varying Br₂ and IBr signals before
t=0 are shown in Fig. S1. Inset more clearly shows the increase of I₂ signal after irradiation. c) NaCl experiment (CL1) at pH = 1.8.
Timescale represents hours from the activation of the lights, and the yellow shading represents presence of radiation from solar
simulator bulbs. Gaps in data represent periods when the isotopic ratios showed an interference.






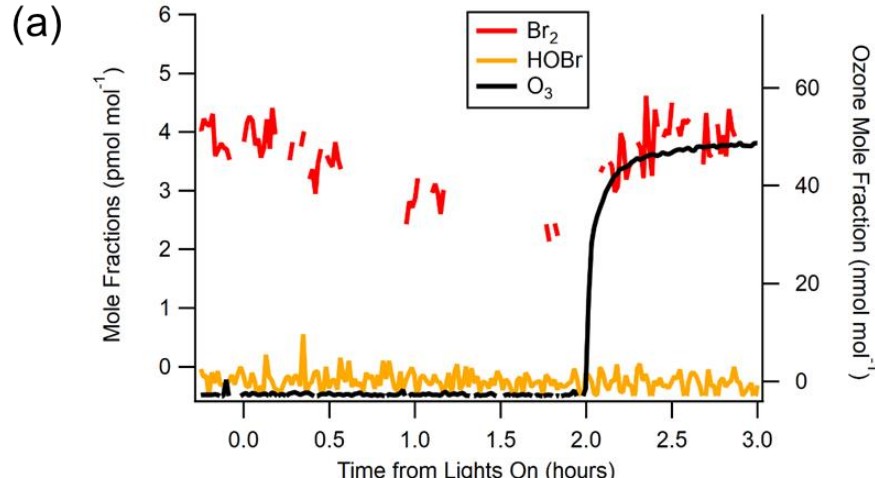

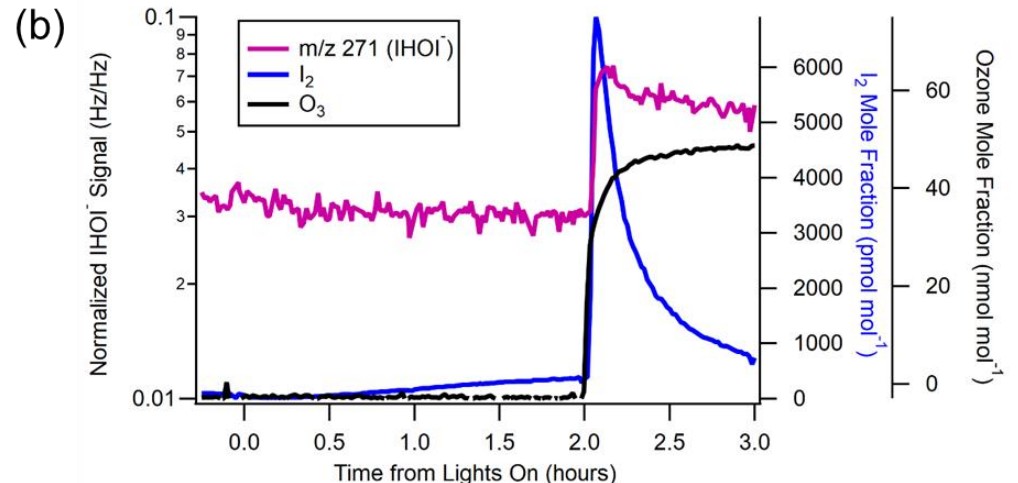


Figure 3: HOX signals from experiment IO2, pH=4.7. a) Comparison of Br₂ mole fractions to HOBr. Note that the HOBr signal, while calibrated, should be used only for qualitative purposes as its identity could not be confirmed using isotopic ratios with *m/z* 223 due to its relatively large background signal. b) Effect of O₃ on I₂ and HOI.




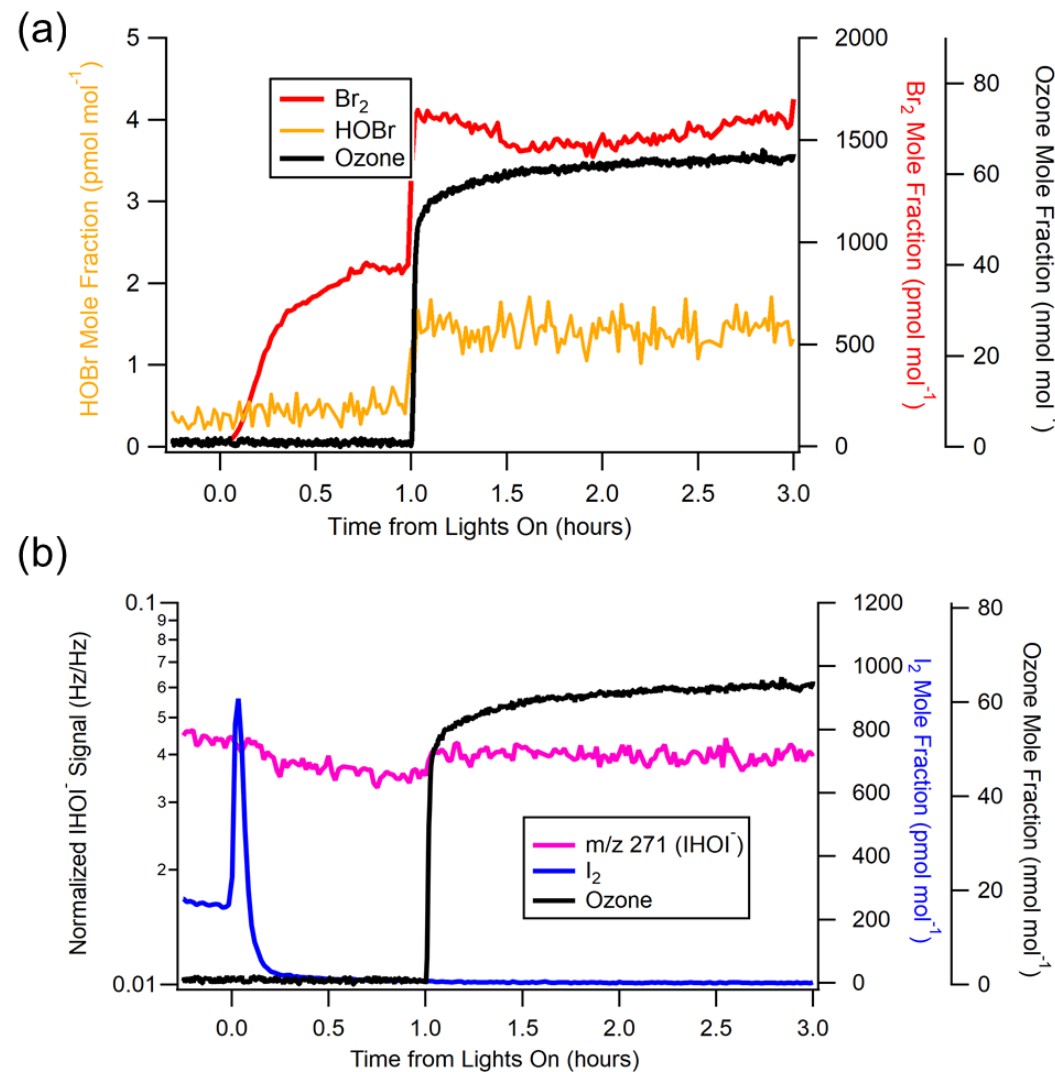


Figure 4: HOX signals from experiment SW5, pH=1.8. a) Comparison of Br$_2$ mole fractions to HOBr (*m/z* 225). Note that the
HOBr signal, while calibrated, should be used only for qualitatively purposes as its identity could not be confirmed using isotopic
ratios with *m/z* 223. b) Effect of O$_3$ on I$_2$ and HOI.