# Peer review of "pH-Dependent production of molecular chlorine, bromine, and 1 iodine from frozen saline surfaces 2"

_Atmospheric Chemistry and Physics, 2018_

## Referee Comment (RC1) · Anonymous Referee #1 · 16 Sep 2018

The manuscript by Halfacre et al. describes a number of laboratory experiments conducted to better elucidate the nature of halogen activation from halide-containing ices. This is an important topic in polar boundary layer chemistry, and only a couple of lab studies have been performed on this system before. So there is inherent merit to the work in that regard.

The halogen release is driven either by exposure to ozone or by illumination when an OH radical precursor, such as nitrite or hydrogen peroxide, is incorporated into the frozen solution. The experiments are performed using a flow tube that is coupled to a chemical ionization mass spectrometer operated with an iodide reagent ion. The

solutions also contain a buffer to maintain the acidity of the solution, given that protons are required for some of these activation processes. I have a few questions concerning the experimental approach (see below) but my major comments are related to the presentation of the experimental results, although I am in general agreement with their interpretations. Some of the results themselves are new and interesting (e.g. I2 release in the dark) but others have been showed before, albeit with different approaches (e.g. Br2 release with light).

I'll start by saying that I found this paper very challenging to read, especially the Results and Discussion sections which were excessively wordy. The Introduction and Conclusions are fine. Given that there are only a few figures of data, I believe that the experimental results and their discussion could be much more succinctly described, perhaps cut in length by a factor of two. As opposed to describing every observation, could the major findings be emphasized? Indeed, I recommend that the authors rewrite the paper so that the results and associated discussion are united, i.e. the results are discussed as they are presented. I found myself moving back and forth many times between the two sections as I was reading the paper.

The Abstract, too, could be improved. For example, the authors should explain what they mean when they state that photochemical production of Br2 was observed (line 33). What were the conditions that led to Br2 production? Another example is that the mechanism of the heterogeneous recycling should be mentioned (line 35). Finally, the last sentence should point out that these mechanisms, even if slow, may be important as the initiation of halogen release to the atmosphere, even if they ultimately do not constitute the major source.

Another weakness of the paper is the attempt to connect the laboratory results to those measured in the field. There are so many factors that come into play in this chemistry, I don't think we can plan to quantitatively relate the lab and the field in the manner attempted. For example, is the spatial distribution of the salts, protons, and OH precursors the same in the field snow samples as those in the lab? I believe the answer

is no, given the totally different manner by which the ice samples are prepared. For example, how might the buffer materials (such as acetate) interact with OH in the ice? Is the surface segregation of these species the same? My advice would be to give up on the goal of making that comparison, except in the most qualitative manner. That said, I do believe that the relative rate approach for interpreting the kinetics of oxidation of different halides has merit. There could be more done interpreting these reactivity ratios in terms of the (much better) known bulk aqueous reaction rate constants.

Overall, I believe there is merit to publication of this paper after the results are presented in a more succinct manner, with caveats added for how this work is translatable to the interpretation of field measurements.

Experimental questions:

Will the acetic acid/acetate buffer be affected by volatilization of acetic acid from the ice?

I found line 128 confusing – i.e. was no iodide observed by IC? If so, what LOD was prevalent for the IC method?

For the bisulfate buffer, will OH react with bisulfate to form the sulfate radical anion, rather than react with the halide ions?

Line 150: Was it room air that went through the flow tube? If so, what contamination may result?

Line 170: What is the spectrum of the solar simulator bulbs?

Line 177: typo

Line 189: Where is the Cl2 background coming from? Just from chloride on flow tube or plumbing surfaces?

Line 203: I am nervous of how the HOBr sensitivity is estimated, given that the CIMS instrument is not the same as used in the referenced work by Liao et al. Wouldn't it be

wiser to just call this signal uncalibrated?

---

## Referee Comment (RC2) · Anonymous Referee #2 · 28 Sep 2018

This study reports experiments related to the photochemical halogen release from frozen sea water mimics. The results provide novel information on the halogen oxidation processes and important basis for the interpretation of field data. It becomes apparent that the most important parameters controlling halogen release are the relative proportions of chloride, bromide and iodide, pH and the structure of the frozen system, in terms of the way in which the combination of crystalline ice and brine are exposed to the gas phase. Since there is also a significant debate in the community on especially the latter two, these aspects could also be illuminated a bit better as detailed below. Overall, the experiments have been carefully designed and analysed, and the analysis of the results is associated with a proper discussion of uncertainties and re-

lated caveats (e.g., related to molecular chlorine detection in presence of ozone). The careful discussion of the halogen cycling reactions and their kinetics is appreciated. I therefore recommend this study for publication with only a few minor suggestions.

Comments:

1) Since the study also tries to differentiate the relative roles of photochemically produced radicals and ozone for halogen release, I wonder why no experiments have been done with ozone only for reference. While such experiments have been done in the past, indeed, exactly because of the complexity of the system in terms of microstructure, the corresponding 'ozone induced baseline dark halogen release' could have been assessed for comparison.

2) the authors several times discuss potential surface reactions occurring on liquid brines, I caution that diffusive exchange even over micrometer ranges is very fast, so that all halide ions present in liquid brine are available for reaction. The kinetics may indeed be limited by a surface process, but this is maybe not the important question, because as observed by the authors, it seems rather that the exchange between compartments may be limiting. If brine in a grain boundary is connected to the surface, diffusion is long enough to allow reaction and release within the experimental time scales. Therefore, the question remains where the less available halide ions are, if brine pockets are probably not buried below ice in such thin films. The way the films were frozen, the ice likely started to grow from the Pyrex glass walls.

3) could the authors please mention more precisely the irradiation conditions and how they were assessed. Has some actinometry been performed?

4) pH: As the authors mention in the experimental part, this is a challenging aspect. I think a short discussion is adequate there and in the discussion section to emphasise the buffer concentrations used in relation to the halide ions, and in what way this may have affected both the physical properties and the halogen / radical chemistry.

---

## Short Comment (SC1) · 15 Oct 2018

T. Bartels-Rausch

thorsten.bartels-rausch@psi.ch

Dear John Halfacre

I read your manuscript with great interest and wish the best for publication in ACP. In particular, I like the increased complexity of your experiments compared to other laboratory studies and the comparison to field data.

Would you mind elaborating in more detail where you think the chemistry is occurring in your samples: the liquid fraction or the ice with its disordered interface? You clearly state that the temperature of of the sample was above the eutectic of NaCl, so we

can expect the presence of liquid in your system. By the way, what the would be the volume of liquid compared to that of ice? Then, later in the discussion the focus is placed on the disordered interface as host of the reactions - as far as I understand the manuscript. I assume you refer to the disordered interface if ice. Could you specify the role of the liquid fraction and of the ice as host for the chemistry? I think at the end this is a semantic issue, as your data are very nicely compared to studies with liquid samples (L. Artiglia, J. Edebeli, F. Orlando, S. Chen, M.-T. Lee, P. Corral Arroyo, A. Gilgen, T. Bartels-Rausch, A. Kleibert, M. Vazdar, M. A. Carignano, J. S. Francisco, P. B. Shepson, I. Gladich and M. Ammann, Nat Comms, 2017, 8, 700.) and to those with frozen samples with a considerable liquid fraction (N. W. Oldridge and J. P. D. Abbatt, J. Phys. Chem. A, 2011, 115, 2590–2598.) Indeed, Oldridge proposed that the reaction occurs in the liquid fraction of their samples.

best regards, Thorsten Bartels-Rausch
* * *

---

## Author Response (AR1)

January 30, 2019

Dear James Roberts,

We have now completed and uploaded responses to the referee comments on our manuscript, "pH-Dependent production of molecular chlorine, bromine, and iodine from frozen saline surfaces," submitted to *Atmospheric Chemistry and Physics.* The referees gave very useful comments, and we feel we have faithfully and completely addressed all the reviewers' concerns. We believe this process has produced a much-improved manuscript, and hope you find this revised paper is now in publishable form.

Sincerely,

John W. Halfacre

**Response to Anonymous Referee #1**

We would like to thank Anonymous Referee #1 for his/her careful reading of this manuscript and constructive suggestions. We have addressed all of his/her comments in the revised manuscript, and describe in detail changes made below in the order in which they were raised by the reviewer. All page numbers and line numbers are in reference to those in the revised version of the manuscript, except where indicated otherwise. For clarity, the text of the reviewer's comments is in **black**, while the authors' responses are in blue.

**The manuscript by Halfacre et al. describes a number of laboratory experiments conducted to better elucidate the nature of halogen activation from halide-containing ices. This is an important topic in polar boundary layer chemistry, and only a couple of lab studies have been performed on this system before. So there is inherent merit to the work in that regard.**

**The halogen release is driven either by exposure to ozone or by illumination when an OH radical precursor, such as nitrite or hydrogen peroxide, is incorporated into the frozen solution. The experiments are performed using a flow tube that is coupled to a chemical ionization mass spectrometer operated with an iodide reagent ion. The solutions also contain a buffer to maintain the acidity of the solution, given that protons are required for some of these activation processes. I have a few questions concerning the experimental approach (see below) but my major comments are related to the presentation of the experimental results, although I am in general agreement with their interpretations. Some of the results themselves are new and interesting (e.g. I2 release in the dark) but others have been showed before, albeit with different approaches (e.g. Br2 release with light).**

**I'll start by saying that I found this paper very challenging to read, especially the Results and Discussion sections which were excessively wordy. The Introduction and Conclusions are fine. Given that there are only a few figures of data, I believe that the experimental results and their discussion could be much more succinctly described, perhaps cut in length by a factor of two. As opposed to describing every observation, could the major findings be emphasized? Indeed, I recommend that the authors rewrite the paper so that the results and associated discussion are united, i.e. the results are discussed as they are presented. I found myself moving back and forth many times between the two sections as I was reading the paper.**

Thank you for your comments and suggestions. We have been diligent in working to improve the readability of the manuscript. The Results and Discussion sections have been combined into Section 3, discussing results as presented, which we believe improved readability, as you suggested. An effort was made to make the new section more concise.

**The Abstract, too, could be improved. For example, the authors should explain what they mean when they state that photochemical production of Br2 was observed (line 33). What were the conditions that led to Br2 production?**

We have amended this sentence to clarify that photochemical production of Br$_2$ at low pH requires an OH precursor (lines 32-33).

**Another example is that the mechanism of the heterogeneous recycling should be mentioned (line 35).**

We have reworded this sentence to clarify that the gas phase HOX compounds would diffuse into our frozen sample solution to oxidize $X^-$ (line 36)

**Finally, the last sentence should point out that these mechanisms, even if slow, may be important as the initiation of halogen release to the atmosphere, even if it is found they do not ultimately constitute the major source.**

We have included this statement at the end of our abstract (lines ~ 40-43), as suggested.

**Another weakness of the paper is the attempt to connect the laboratory results to those measured in the field. There are so many factors that come into play in this chemistry, I don't think we can plan to quantitatively relate the lab and the field in the manner attempted. For example, is the spatial distribution of the salts, protons, and OH precursors the same in the field snow samples as those in the lab? I believe the answer is no, given the totally different manner by which the ice samples are prepared. For example, how might the buffer materials (such as acetate) interact with OH in the ice? Is the surface segregation of these species the same? My advice would be to give up on the goal of making that comparison, except in the most qualitative manner. My advice would be to give up on the goal of making that comparison, except in the most qualitative manner.**

We have reevaluated our discussion points as a result of this comment and have removed, most notably, the calculation of relative production rates for the field based on our calculated relative reactivities. While we feel the calculation of relative reactivities represents an important empirical result, it is less defensible to apply them quantitatively to field observations derived from samples very different from ours, as the reviewer suggests.

Interactions between buffers and OH are discussed in more detail below under the more specific question from the reviewer. We have also incorporated discussion to this end in the Supplemental Information, lines 3-28.

The reviewer is also correct in implying vast differences in the spatial distribution of ions between our lab samples and field samples. We acknowledge and discuss this point in the paragraph that spans lines 386-407 in the revised manuscript.

**That said, I do believe that the relative rate approach for interpreting the kinetics of oxidation of different halides has merit. There could be more done interpreting these reactivity ratios in terms of the (much better) known bulk aqueous reaction rate constants.**

A comparison of our calculated relative reactivities with the bulk aqueous OH-halide rate constants, along with a discussion of the implications of the differences, has been included and begins on line 372.

**Experimental questions-**

**Will the acetic acid/acetate buffer be affected by volatilization of acetic acid from the ice?**

pH measurements were recorded before and after each experiment, but no statistically significant difference was observed. We have now clarified that this suggests no significant loss of the buffering capacity over the course of the experiment on lines 146-148.

**I found line 128 confusing – i.e. was no iodide observed by IC? If so, what LOD was prevalent for the IC method?**

Iodide was not observed above the observed limit of detection by IC. We have clarified this point and added the limit of detection for the method used (90 nM), line 327.

**For the bisulfate buffer, will OH react with bisulfate to form the sulfate radical anion, rather than react with the halide ions?**

This reaction pathway will indeed occur, potentially followed by reaction with the halides.

$$\cdot OH + HSO_4^- \longrightarrow H_2O + SO_4\cdot^- \qquad\qquad k = 4.7 \times 10^5 \text{ M}^{-1} \text{ sec}^{-1}$$

However, the OH reactions rates with the halides are >10,000 faster ($k_{Cl^-} = 3 \times 10^9$ M$^{-1}$ sec$^{-1}$, $k_{Br^-} = k_{I^-} = 1.1 \times 10^{10}$ M$^{-1}$ sec$^{-1}$). We calculate that the sulfate radical would contribute less than 0.1% to halide oxidation, compared that from OH-halide oxidation. Because this is only believed to influence the results to a minor degree, we have included this discussion in the Supplemental Information, lines 3-28.

**Line 150: Was it room air that went through the flow tube? If so, what contamination may result?**

The flow tube was cleaned in between experiments with three rinses of ultrapure water with a final rinse of acetone. It was then dried with a > 99.99% purity nitrogen cylinder before being capped. Likely a small amount of room air that contains small amounts of ozone and nitrogen oxides diffuses into the flow tube during the addition of sample to the tube. However, this air would be quickly flushed from the flow tube by zero air on connection with the CIMS, and experimental data are only obtained after signals stabilize. We thank the reviewer for raising this point, and now discuss it on lines 154 as well as detail our cleaning procedure on lines ~181-185.

**Line 170: What is the spectrum of the solar simulator bulbs?**

We have included the solar spectrum of the solar simulator bulbs in Figure S1.

**Line 177: typo**

"Ion" was corrected to "Ions"

**Line 189: Where is the Cl2 background coming from? Just from chloride on flow tube or plumbing surfaces?**

The signal at $m/z$ 197 during ozone addition is not due to $Cl_2$ because corresponding isotopic signals at m/z 199 and 201 did not rise in concert. This has been clarified in Sect. 2.3, lines 201-203.

**Line 203: I am nervous of how the HOBr sensitivity is estimated, given that the CIMS instrument is not the same as used in the referenced work by Liao et al. Wouldn't it be wiser to just call this signal uncalibrated?**

On line 203-206 of the revisions, we have indicated that the HOBr signal is uncalibrated, and only discuss the relative changes in HOBr signal. Figures 3 and 4 have been adjusted to present the $IHOBr^-$ signals instead of the estimated mole fractions.

Response to Anonymous Referee #2
We would like to thank Anonymous Referee #2 for his/her careful reading of this manuscript and constructive suggestions. We have addressed all of his/her comments in the revised manuscript, and describe in detail changes made below in the order in which they were raised by the reviewer. All page numbers and line numbers are in reference to those in the revised version of the manuscript, except where indicated otherwise. For clarity, the text of the reviewer's comments are in **black**, while the authors' responses are in blue.

**This study reports experiments related to the photochemical halogen release from frozen sea water mimics. The results provide novel information on the halogen oxidation processes and important basis for the interpretation of field data. It becomes apparent that the most important parameters controlling halogen release are the relative proportions of chloride, bromide and iodide, pH and the structure of the frozen system, in terms of the way in which the combination of crystalline ice and brine are exposed to the gas phase. Since there is also a significant debate in the community on especially the latter two, these aspects could also be illuminated a bit better as detailed below. Overall, the experiments have been carefully designed and analysed, and the analysis of the results is associated with a proper discussion of uncertainties and related caveats (e.g., related to molecular chlorine detection in presence of ozone). The careful discussion of the halogen cycling reactions and their kinetics is appreciated. I therefore recommend this study for publication with only a few minor suggestions.**

**Comments:**

**1) Since the study also tries to differentiate the relative roles of photochemically produced radicals and ozone for halogen release, I wonder why no experiments have been done with ozone only for reference. While such experiments have been done in the past, indeed, exactly because of the complexity of the system in terms of microstructure, the corresponding 'ozone induced baseline dark halogen release' could have been assessed for comparison.**

Our experiments were inspired by the field work of Pratt et al. (2013) and Raso et al. (2017) that suggested evidence of photochemically-initiated halogen production, the testing of which was the primary focus of this paper. We appreciate the author's comment and acknowledge that such dark experiments with ozone only would have allowed for a more direct comparison of our results with previous laboratory experiments, such as Oum et al. (1998) and Oldridge and Abbatt (2011), discussed in our paper on lines 102-106. However, since Pratt et al. (2013) demonstrated that $O_3 + Br^-$ was relatively unimportant as a $Br_2$ source under normal atmospheric $O_3$ conditions without radiation, we focused on the role of OH as an initiator.

**2) The authors several times discuss potential surface reactions occurring on liquid brines, I caution that diffusive exchange even over micrometer ranges is very fast, so that all halide ions present in liquid brine are available for reaction. The kinetics may indeed be limited by a surface process, but this is maybe not the important question, because as observed by the authors, it seems rather that the exchange between**

**compartments may be limiting. If brine in a grain boundary is connected to the surface, diffusion is long enough to allow reaction and release within the experimental time scales. Therefore, the question remains where the less available halide ions are, if brine pockets are probably not buried below ice in such thin films. The way the films were frozen, the ice likely started to grow from the Pyrex glass walls.**

We thank the Referee for this comment. The recent work from Malley et al. (2018), cited line 388) discusses the brine distribution in frozen surfaces in much more depth, and we look forward to experiments inspired by this work that will provide further clarity on this issue. But, we expect that the less-available ions are locked in the bulk ice, as discussed in lines 386-407 of the revision.

**3) Could the authors please mention more precisely the irradiation conditions and how they were assessed? Has some actinometry been performed?**

We have reproduced the solar irradiance spectrum of the solar simulator bulbs in Figure S1. No actinometry was performed, but we do not attempt in this paper to simulate the actual ambient radiant fluxes, but rather discuss the relative rates of production, and the roles of pH, and OH and $O_3$ in the gas phase on those rates.

**4) pH: As the authors mention in the experimental part, this is a challenging aspect. I think a short discussion is adequate there and in the discussion section to emphasize the buffer concentrations used in relation to the halide ions, and in what way this may have affected both the physical properties and the halogen / radical chemistry.**

The hydroxyl radical can react with acetic acid, as well as with bisulfate to form sulfate radical:

$$\cdot OH + HSO_4^- \longrightarrow H_2O + SO_4 \cdot^- \qquad\qquad k = 4.7 \times 10^5 \ M^{-1} \ sec^{-1}$$

$$CH_3CO_2H + \cdot OH \longrightarrow H_2O + \cdot CH_2CO_2H \qquad k = 9.2 \times 10^6 \ M^{-1} \ sec^{-1}$$

However, the OH reactions rates with the halides are considerably faster ($k_{Cl^-} = 3 \times 10^9 \ M^{-1} \ sec^{-1}$, $k_{Br^-} = k_{I^-} = 1.1 \times 10^{10} \ M^{-1} \ sec^{-1}$). We calculate that the sulfate radical would contribute less than 0.1% to halide oxidation, compared that from OH-halide oxidation. Because this is only believed to influence the results to a minor degree, we have included the desired discussion in the Supplemental Information, Lines 3-28.

**Response to Dr. Bartles-Rausch**
We would like to thank Dr. Bartels-Rausch for his interest in our manuscript and for his comments. We have addressed his comments/questions below in the order in which they were raised. All page numbers and line numbers are in reference to those in the revised version of the manuscript, except where indicated otherwise. For clarity, the text of the Dr. Bartles-Rausch comments is in **black**, while the authors' responses are in blue.

**I read your manuscript with great interest and wish the best for publication in ACP. In particular, I like the increased complexity of your experiments compared to other laboratory studies and the comparison to field data.**

**Would you mind elaborating in more detail where you think the chemistry is occurring in your samples: the liquid fraction or the ice with its disordered interface? You clearly state that the temperature of the sample was above the eutectic of NaCl, so we can expect the presence of liquid in your system.**

Based on the temperature of the experiments, and the ionic strength of the water samples use to make the ice coating, we believe the reactions occur primarily in a liquid brine on the surface of the ice layer and have clarified on lines 158-163, as well as line 2323 of the revised manuscript. This is consistent with Cho et al. (2001) and Oldridge and Abbatt (2011).

**By the way, what would be the volume of liquid compared to that of ice?**

Our experimental sample was 80.0 mL of an Instant Ocean solution that was made to be approximately 0.56 M with respect to NaCl (the most abundant ions in the salt). Based on simple freezing point depression thermodynamics for a 0.56M NaCl solution and an ice T = -15°C, we calculate a liquid fraction of 0.124.

**Then, later in the discussion the focus is placed on the disordered interface as host of the reactions - as far as I understand the manuscript. I assume you refer to the disordered interface if ice. Could you specify the role of the liquid fraction and of the ice as host for the chemistry? I think at the end this is a semantic issue, as your data are very nicely compared to studies with liquid samples (L. Artiglia, J. Edebeli, F. Orlando, S. Chen, M.-T. Lee, P. Corral Arroyo, A. Gilgen, T. Bartels-Rausch, A. Kleibert, M. Vazdar, M. A. Carignano, J. S. Francisco, P. B. Shepson, I. Gladich and M. Ammann, Nat Comms, 2017, 8, 700.) and to those with frozen samples with a considerable liquid fraction (N. W. Oldridge and J. P. D. Abbatt, J. Phys. Chem. A, 2011, 115, 2590–2598.) Indeed, Oldridge proposed that the reaction occurs in the liquid fraction of their samples.**

For our analysis, we assumed that all of the chemistry occurred in the brine, as indicated on lines 158-163 and 232 of the revised manuscript.

[revised manuscript text omitted]

**2. Methods**

**2.1    Materials**

Acetic acid/acetate and bisulfate/sulfate buffer concentrations were 20 mM (10 mM of each acid and conjugate base).  This concentration was chosen as a compromise between using as little buffer as possible and enough buffer to ensure adequate buffering ability, as buffer capacity rapidly decreases as constituent species concentrations approach the acid $K_a$ value. The halide concentrations from our salt water solutions were $Cl^-$ 500mM, $Br^-$ 0.72mM, and $I^-$ $1.9 \times 10^{-3}$ mM.

Given that the buffer concentration is comparable to or exceeds halide ion concentrations, there is a concern that buffer composition may change over time due to the volatility of acetic acid (Henry's Law Constant of 400 M/atm), or because of buffer reactions with OH that may compete with reactions between OH and the halides:

$\cdot OH + HSO_4^- \longrightarrow H_2O + SO_4 \cdot^-$            $k = 4.7 \times 10^5$ $M^{-1}$ $sec^{-1}$

$\underline{Cl^-} + SO_4 \cdot^- \longrightarrow \underline{Cl \cdot} + SO_4^{2-}$            $k = 2.6 \times 10^8$ $M^{-1}$ $sec^{-1}$

$\underline{Br^-} + SO_4 \cdot^- \longrightarrow SO_4^{2-} + \underline{Br \cdot}$            $k = 3.5 \times 10^9$ $M^{-1}$ $sec^{-1}$

$CH_3CO_2H + \cdot OH \rightarrow H_2O + \cdot CH_2CO_2H$        $k = 9.2 \times 10^6$ $M^{-1}$ $sec^{-1}$

Using these aqueous rate constants and the pre-freezing concentrations of species in our paper, we find the following relative rates of OH-based production:

$\frac{\frac{d[X_2]}{dt}}{\frac{d[SO_4]}{dt}} = 3.6 \times 10^5, 1.7 \times 10^3, 4.4$ for $Cl_2$, $Br_2$, and $I_2$, respectively.

$\frac{\frac{d[X_2]}{dt}}{\frac{d[CH3CO2H]}{dt}} = 1.8 \times 10^4, 8.6 \times 10^1, 2.3 \times 10^{-1}$ for $Cl_2$, $Br_2$, and $I_2$, respectively.

It is clear based on these relative rates of production that sulfate radical may contribute only a minor amount of $Br^-$ and $Cl^-$ oxidation in our experiments, less than 0.1% of that from OH-halide oxidation.

No comparable rate constant could be found between $I^-$ and sulfate.  $I_2$ production may be impacted by competition of the $HSO_4^-$ and OH.  Including dark production, however, $I_2$ was consistently our most abundant product in all experiments except CL1 (in which only trace, undetectable iodide may have been present).  Further, we do not anticipate them occurring to an appreciable degree based on the fact that pH measurements before and after experiments were identical (indicating no significant depletion of either buffer species throughout the experiment).

**2.2    Flow tube**

Reaction photochemistry was achieved using six UVA-340 solar simulator lamps (Q-Labs, 295 – 400 nm with maximum wattage at 340 nm, irradiance spectrum in Fig. S1). These lamps were installed in the experiment box (two on each side, except bottom). Each side was lined with reflective Mylar sheets to evenly irradiate the flow tube when the lamps were powered.

**2.3    CIMS**

Experiments utilizing the bisulfate/sulfate buffer (IO3-5, IO8, SW3-5, SW8, and CL1) sometimes exhibited cyclical CIMS signal changes for $Br_2$ (*m/z* 285, 287, 291), IBr (*m/z* 333, 335) with no attributable cause. These signal changes occurred seemingly at random and to varying extents.  In Fig. S2a, Experiment IO4 (pH = 1.7, includes $H_2O_2$) demonstrates the most extreme example of this behaviour that almost appears to affect the analysis.  First at t = -3, the $Br_2$ rises briefly before falling. Then at t=2, the $Br_2$ signal begins to resemble a sine wave. All data beyond t=2 is not considered for this specific experiment.  In Fig S2b, the effect during Experiment SW5 (pH = 1.7, includes $H_2O_2$) is more muted, beginning at approximately t = -6 for IBr and $Br_2$.  As represented by these figures, this behaviour being farther away from our periods of integration is typical of the remaining experiments. Because these signal changes occurred outside of the experimental periods analyzed (i.e., before irradiation, and after $O_3$ had been active for one hour), they are therefore not believed to affect our results and their interpretation.

**3  Results and Discussion**

**3.1    Dark reaction production of $I_2$**

In cases without OH precursors at pH < 2, significant photochemical $I_2$ production still occurs (integrated production of 14 ± 10 nmol for IO8, and 6.0 ± 2.0 nmol for SW8), while $Br_2$ and $Cl_2$ concentrations remain below limits of detection (consistent with Abbatt et al., (2010), in which no $Br_2$ was observed without an OH-precursor) (Table 2, main text).  This production likely stems from the mechanisms outlined by Kim et al. (2016) (R13-14, R10-R12), discussed in the Sect. 1.  As discussed in Sect. 3.1, $H_2O_2$ or $NO_2^-$ can react directly with $I^-$, thereby reducing the available $[I^-]$ for photochemical OH oxidation when pH < 2.  When $H_2O_2$ was the oxidant, integrated $I_2$ production amounts were found to be $\leq 0.82$ nmol (IO4, IO5, and SW5), likely due to this initial dark depletion. When instead

$NO_2^-$ is used (as in IO3 and SW3), initial amounts of $I_2$ on flowtube connection to CIMS were less than when $H_2O_2$

was used (Table S1, Fig. S3). To estimate how much $I^-$ may have been lost from our frozen sample by these dark mechanisms, we convert the integrated $I_2$ production amounts from Table S1 to $I^-$ (by multiplying by 2) and subtract from the maximum possible moles of $I^-$ in our samples (0.0800 L * 1.6 x $10^{-6}$ M = 1.28 x $10^{-7}$ moles $I^-$). For the samples that use hydrogen peroxide, as little as 36– 91% of $I^-$ is available for reaction, while 94-97% remain when using $NO_2^-$. However, it is certain that not all of the $I_2$ produced by this mechanism went into the CIMS by the nature of having to break the flow tube seal in order to connect it to the CIMS. Therefore, these are only estimates that could be affected by the length of time the tube is open to the environment and not connected to the CIMS, or sealed shut.

**63    3.2 Hydroxyl radical-induced halogen production**

**64    3.2.1 pH $\approx 4.7$**

Considering the values of $I_2$ production from Table 2 (main text), IO2, appears to have produced ~10 times less $I_2$ based on the chosen period of integration. It was noted that $I_2$ appeared to already be present within the flow tube on connecting the flow tube to the CIMS (Fig. S4). The integrated sum of $I_2$ released on connection of the flow tube to the CIMS until stabilization was 0.8 ($\pm 0.1$) nmol, corresponding to approximately 0.5% of the total 152 nmol

$I^-$ available for reaction from the Instant Ocean solution (Table S1). This production could possibly be induced by the dark reactions described in Sect. 3.1. However, the experiment otherwise eventually produces the same qualitative features as the other three experiments after light activation (Fig. S4). If instead the limits of integration are chosen starting when the $I_2$ signal begins rising (i.e., during a period that qualitatively resembles the other experiments), the integrated $I_2$ production amounts ($1.1 \pm 0.6$ nmol) more closely approaches analogous experiments (IO1, SW1, SW2).

The apparent photochemical integrated $Br_2$ sum of $0.034 \pm 0.003$ nmol (Table 2) represents a real signal just above the limit of detection ($1.8 \pm 0.4$ pmol $mol^{-1}$), but this baseline signal does not change on addition of light (Fig. 3a). In addition, the integration method used likely interpolated missing data for time periods in which incorrect isotope ratios between $m/z$ 285 and 287 were observed, thereby overestimating the integrated yield. This signal remains below limits of quantitation and should not be considered further. $Cl_2$ concentrations remained below limits of detection for experiment IO2.

[revised manuscript text omitted]

---

## Author Response (AR2)

March 1, 2019

Dear James Roberts,

We have responded below to your specific comments regarding the supplement to our manuscript, "pH-Dependent production of molecular chlorine, bromine, and iodine from frozen saline surfaces," submitted to *Atmospheric Chemistry and Physics.* We thank you for your careful attention to this section, and we feel we have successfully addressed and clarified your questions, as indicated below (your comments in black, our responses in blue). We have also included in this document the revised Supplement with changes highlighted in yellow. We hope you find the manuscript is now complete and in publishable form.

Sincerely,

John W. Halfacre

I think you have answered all the reviewers' issues. I just have a few of my own.

Lines 15-20 in the Supplemental need some explanation and the first rate equation needs some work. Since you are using d[X2]/dt, you are essentially assuming the OH + Cl- is the rate limiting step to forming X2. This needs to be explicitly stated here.

This passage has been clarified to indicate we are comparing the relative rates with which OH directly reacts with either the halides, or the buffer constituents ($HSO_4^-$ or acetic acid) (lines 20-21). We have additionally clarified by altering "$d[X_2]/dt$" to "$d[X^-]/dt$" (lines 22-23).

Also, I don't understand d[SO4]/dt. Shouldn't it be d[HSO4-]? and doesn't this also assume that OH + HSO4- is the rate limiting step here? This should also be stated here.

You are correct, and we have adjusted the equation (line 22). We have additionally clarified that we are comparing competition between the direct reactions of $X^-$ and buffer constituents ($HSO_4^-$ and acetic acid) with OH, and not the $X_2$ rate of formation (lines 20-21).

**2. Methods**

**2.1    Materials**

Acetic acid/acetate and bisulfate/sulfate buffer concentrations were 20 mM (10 mM of each acid and conjugate base).  This concentration was chosen as a compromise between using as little buffer as possible and enough buffer to ensure adequate buffering ability, as buffer capacity rapidly decreases as constituent species concentrations approach the acid $K_a$ value. The halide concentrations from our salt water solutions were Cl⁻ 500mM, Br⁻ 0.72mM, and I⁻ $1.9 \times 10^{-3}$ mM.

Given that the buffer concentration is comparable to or exceeds halide ion concentrations, there is a question of whether buffer composition may change over time due to the volatility of acetic acid (Henry's Law constant of 400 M/atm), or because of buffer-constituent reactions with OH (concentration of 100 mM) that may compete with reactions between OH and halides. Here we present these potential reactions, associated rate constants, and calculate the potential for artifacts due to the presence of the buffer.

$\cdot OH + I^- \rightarrow HOI^-$            $k = 1.1 \times 10^{10}\ M^{-1}\ sec^{-1}$ (Buxton et al., 1988)

$\cdot OH + Br^- \rightarrow HOBr^-$        $k = 1.1 \times 10^{10}\ M^{-1}\ sec^{-1}$ (Zehavi and Rabani, 1972)

$\cdot OH + Cl^- \rightarrow HOCl^-$         $k = 3.0 \times 10^{9}\ M^{-1}\ sec^{-1}$ (Grigor'ev et al., 1987)

$CH_3CO_2H + \cdot OH \rightarrow H_2O + \cdot CH_2CO_2H$    $k = 9.2 \times 10^{6}\ M^{-1}\ sec^{-1}$ (Thomas, 1965)

$\cdot OH + HSO_4^- \rightarrow H_2O + SO_4 \cdot^-$     $k = 4.7 \times 10^{5}\ M^{-1}\ sec^{-1}$ (Jiang et al., 1992)

$Cl^- + SO_4 \cdot^- \rightarrow Cl \cdot + SO_4^{2-}$       $k = 2.6 \times 10^{8}\ M^{-1}\ sec^{-1}$ (Padmaja et al., 1993)

$Br^- + SO_4 \cdot^- \rightarrow SO_4^{2-} + Br \cdot$       $k = 3.5 \times 10^{9}\ M^{-1}\ sec^{-1}$ (Redpath and Willson, 1975)

Using these aqueous rate constants and the pre-freezing concentrations of species (above and in main text Sect. 2.1), we find the following relative rates of OH reactions with halides, compared to OH reactions with buffer constituents:

$$\frac{\frac{d[X^-]}{dt}}{\frac{d[HSO_4^-]}{dt}} = 3.6 \times 10^5, 1.7 \times 10^3, \text{and } 4.4 \text{ for Cl}^-, \text{Br}^-, \text{and I}^-, \text{respectively.}$$

$$\frac{\frac{d[X^-]}{dt}}{\frac{d[CH_3CO_2H]}{dt}} = 1.8 \times 10^4, 8.6 \times 10^1, 2.3 \times 10^{-1} \text{ for Cl}^-, \text{Br}^-, \text{and I}^-, \text{respectively.}$$

It is clear based on these relative rates of production that, assuming OH + HSO₄⁻ is rate limiting, sulfate radical production would contribute only minorly to Br⁻ and Cl⁻ oxidation in our experiments, i.e., less than 0.1% of that from OH-halide oxidation.  No analogous rate constant could be found for I⁻ reaction with the sulfate radical anion, and so it is unclear the extent to which I⁻ oxidation (and subsequent $I_2$ formation) may be impacted by formation of the sulfate radical. The oxidation of acetic acid has no impact on our experiments to our knowledge. While it might decrease the OH radical concentration, this would not impact our study of the relative rates of halide oxidation by OH. This is further supported by the fact that the pH measurements before and after experiments were statistically identical (indicating no significant depletion of either buffer species throughout the experiment, and consequently, no significant depletion of OH by reactions with either buffer species).

**2.2 Flow tube**

[revised manuscript text omitted]